# Revised oceanic molybdenum isotope budget from deep-sea pelagic sediments

Zhibing Wang [1,2] ✉, Jie Li[1], Bangqi Hu[3], Liang Zou [3] ✉, Xue Ding[3], Le Zhang [1], Jinlong Ma[1] & Gangjian Wei[1,2]

Molybdenum isotopes serve as critical proxies for reconstructing ancient ocean oxygenation, yet the modern oceanic Mo isotopic budget remains incompletely understood. Deep-sea pelagic sediments enriched in Fe-Mn (hydro)oxides represent a major oxic sink, but their authigenic Mo isotopic composition is poorly constrained. Here, we show Mo isotope data from Pacific deep-sea sediment cores revealing systematic depth-dependent $\delta^{98}$Mo enrichment from –0.55‰ to 0.19‰, controlled by Fe-Mn cycling during early diagenesis. Combined with existing datasets, we calculate a revised authigenic oxic Mo flux of $1.52 \times 10^8$ mol yr$^{-1}$ with $\delta^{98}$Mo = –0.09 ± 0.23‰—more than double previous estimates and ~0.6‰ heavier than Fe-Mn crusts. These findings necessitate recalibration of the global Mo isotope budget and demonstrate that pelagic sediments exert greater influence on oceanic Mo cycling than previously recognized with implications for quantitative paleoceanographic reconstructions.

Molybdenum (Mo) serves as a powerful proxy for reconstructing paleoceanographic redox conditions through its abundance and isotopic variations[1–9]. A precise understanding of the modern oceanic Mo cycle and its isotopic budget is therefore critical for validating Mo-based paleoredox interpretations[1–3,8]. Despite extensive research characterizing the Mo isotopic composition of major oceanic sources and sinks[1,8,10], significant uncertainties remain in the global Mo budget. Current models assume that Fe-Mn crusts and nodules accurately represent the isotopic signature of all marine Fe-Mn (hydro)oxides sediments[8]. This assumption, however, overlooks the substantial reservoir of dispersed Fe-Mn (hydro)oxide within pelagic sediments, which far exceeds the volume of crusts and nodules[11,12]. These pelagic sediments, characterized by Fe-Mn (hydro)oxide grain coatings and micronodules, constitute a major oxic Mo sink[1,13,14]. The disparity between these reservoir sizes challenges the validity of using crusts and nodules as representative of the entire Fe-Mn (hydro)oxides sink. Moreover, the isotopic signatures of metals adsorbed by Fe-Mn crusts and nodules may differ from those adsorbed by Fe-Mn (hydro)oxides particles in pelagic sediments, potentially leading to variations in authigenic $\delta^{98}$Mo values[15,16]. To address these uncertainties and

advance our understanding of the global oceanic Mo isotope budget, a comprehensive investigation of Mo isotopic compositions in Fe-Mn (hydro) oxides-rich deep-sea pelagic sediments is necessary. In this study, we analyzed the Mo isotopic compositions of two deep-sea pelagic sediment columns collected from the western Pacific Ocean (Fig. 1). By synthesizing these data with existing Mo concentration and isotopic measurements from deep-sea pelagic sediments, we derive a revised authigenic oxic Mo flux of $1.52 \times 10^8$ mol yr$^{-1}$ with a $\delta^{98}$Mo value of –0.09 ± 0.23‰. These findings propose a revised global Mo isotope budget, advancing understanding of pelagic sediment contributions and improving the precision of Mo isotope-based paleoceanographic reconstructions. The updated global Mo isotope mass balance model suggests that previous studies significantly overestimated the extent of euxinic seafloor in ancient oceans.

## Results and discussion

### Mo Isotope variations in deep-sea pelagic sediment

The geochemical and Mo isotopic profiles exhibit systematic variations with depth in both sediment cores (Supplementary Table S1). Titanium (Ti), a reliable indicator for quantifying detrital input in

[1]State Key Laboratory of Deep Earth Processes and Resources, Guangzhou Institute of Geochemistry, Chinese Academy of Sciences, Guangzhou, China. [2]College of Earth and Planetary Sciences, University of Chinese Academy of Sciences, Beijing, China. [3]Qingdao Institute of Marine Geology, China Geological Survey, Qingdao, China. ✉e-mail: wangzhibing@gig.ac.cn; zouliang04@163.com

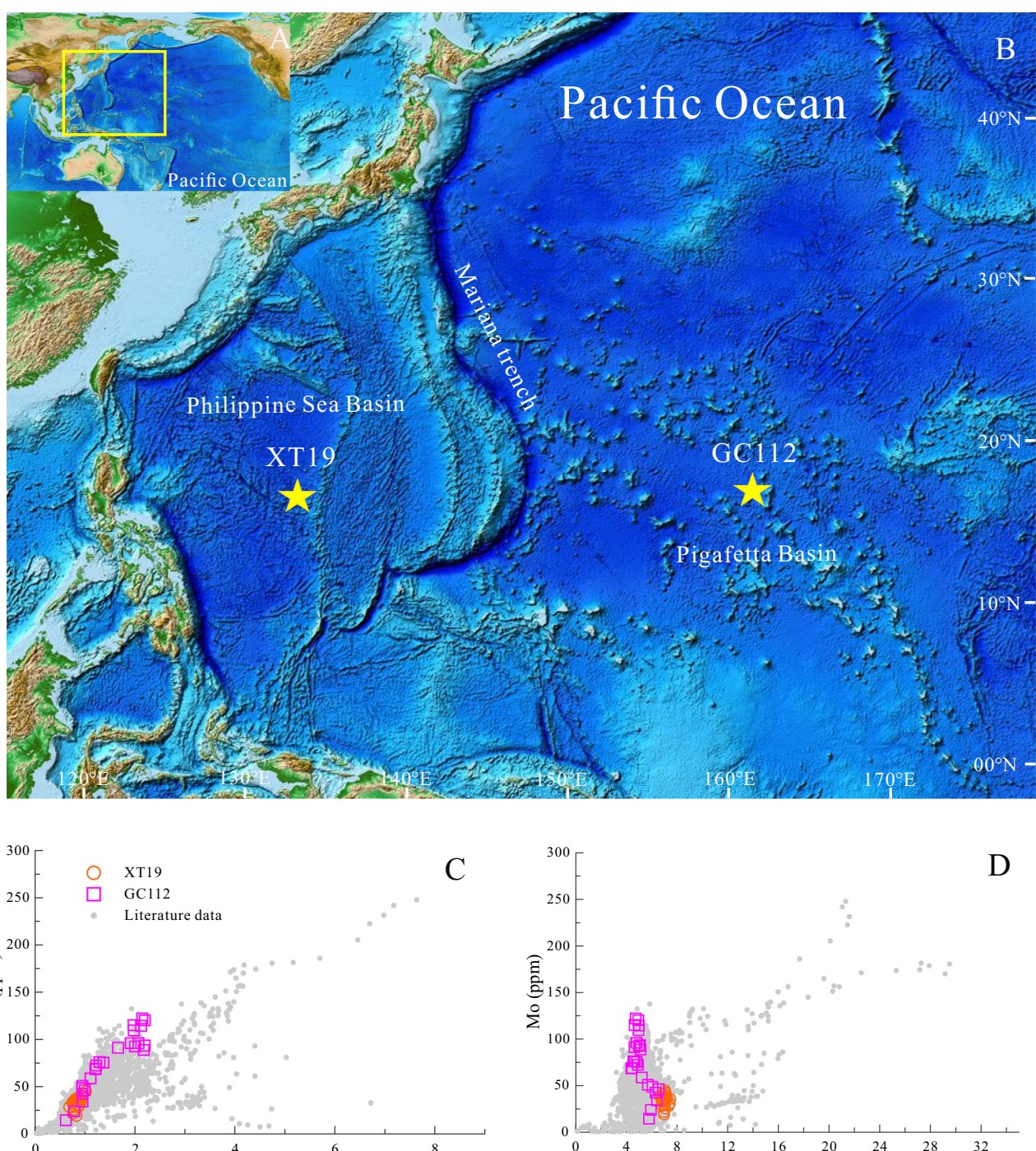

**Fig. 1 | Geochemical characteristics and sampling locations of the western Pacific deep-sea sediments. A, B** Bathymetric maps displaying the sampling locations (yellow stars). The base map is derived from the ETOPO1 1-arc-minute Global Relief Model[67]. **C, D** Cross-plots showing the relationship between bulk-sediment Mo concentration and that of Mn and Fe. For comparative analysis, a comprehensive dataset ($n = 1955$) from both the Indian and Pacific Oceans is included. The complete dataset is available in the Supplementary Dataset 1, 2 hosted on Mendeley Data[66].

marine sediments, is employed to normalize Mn, Fe, and Mo concentrations (see Supplementary Information S1 for further details). The Mn/Ti, Fe/Ti, and Mo/Ti ratios exhibit a progressive increase with depth from 0 to 4.0 mbsf, remaining stable in the deeper layers of core GC112, while showing a slight upward trend in core XT19 (Fig. 2A–C). Conversely, the Mn/Mo and Fe/Mo ratios demonstrate consistent decreases throughout both profiles (Fig. 2E, F). The $\delta^{98}$Mo values in these deep-sea sediments, ranging from –0.55‰ to 0.19‰ (Fig. 2D),

within the range of deep-sea sediment reported in previous studies[15–17], are significantly higher than those reported for Fe-Mn nodules and crusts (–0.70‰)[1,18]. Core XT19 exhibits a general increase in $\delta^{98}$Mo values from –0.52 ± 0.04‰ to 0.12 ± 0.08‰, characterized by an initial rise at ‑1.69 mbsf followed by a slight decline. Overall, $\delta^{98}$Mo values in core GC112 exhibit a gradual increase with depth, ranging from –0.55 ± 0.04‰ to 0.19 ± 0.03‰, interspersed with minor decreases at several depths (e.g., 2.5 mbsf, 4.2 mbsf, and 6.0 mbsf). This vertical

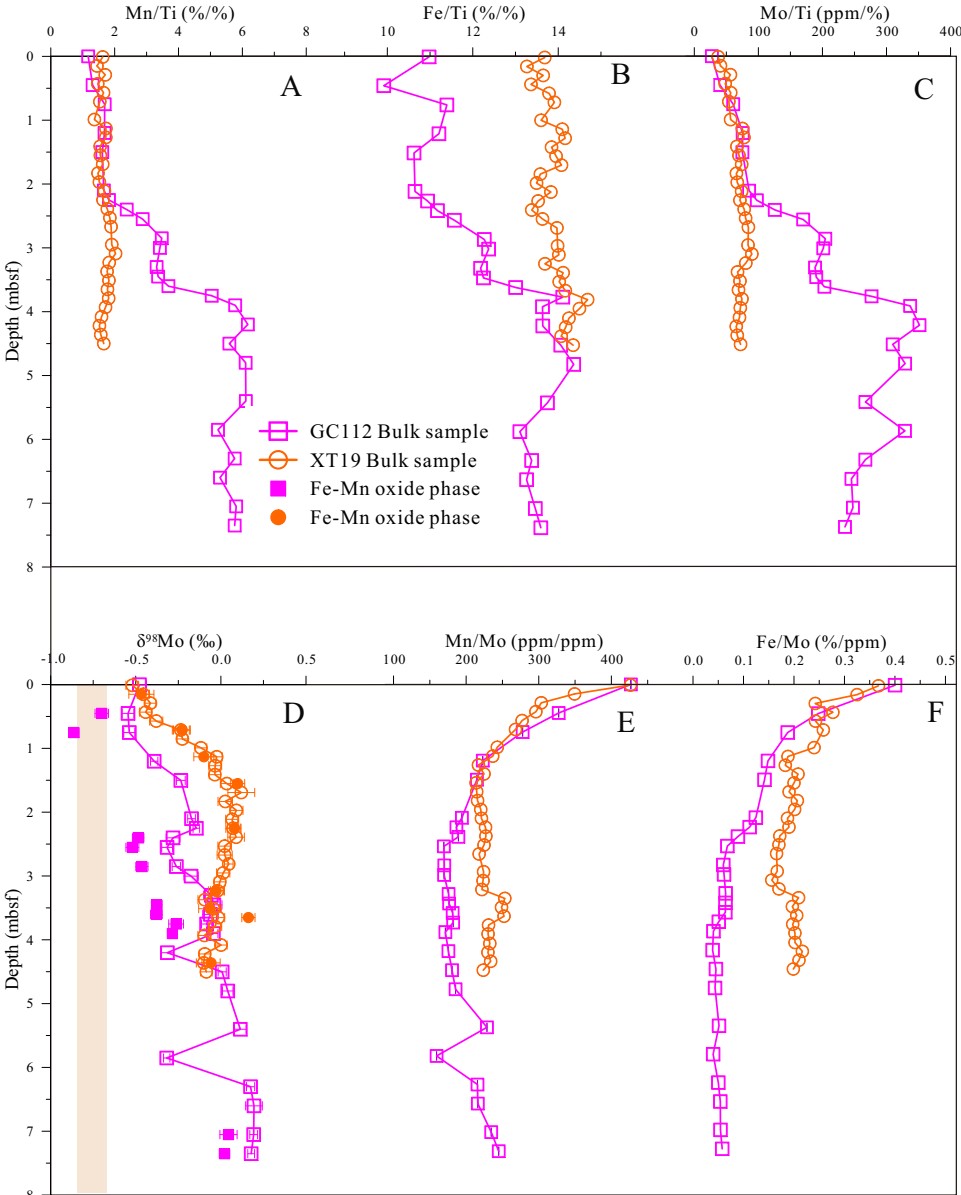

**Fig. 2 | Depth-dependent Mo isotopes and metal ratios in western Pacific deep-sea sediments.** The plots show depth profiles for: (**A**) Mn/Ti, (**B**) Fe/Ti, (**C**) Mo/Ti, (**E**) Fe/Mo, and (**F**) Mn/Mo in bulk samples. **D** The $\delta^{98}$Mo values are shown for both bulk sediments (hollow symbols) and their extracted Fe-Mn (hydro)oxide phases (filled symbols). "mbsf" denotes meters below the seafloor. For reference, the yellow bar represents the average $\delta^{98}$Mo of hydrogenetic Fe-Mn crusts and nodules ($-0.70 \pm 0.14‰$)[1].

pattern parallels observations from the Central North Pacific Ocean, where $\delta^{98}$Mo values increase from $-0.75 \pm 0.05‰$ at the sediment-water interface to $1.0 \pm 0.03‰$ at 3.0 mbsf, and then decrease to 0.50‰ at the bottom layer (>3.0 mbsf)[17]. Similarly, Mo isotopes in the deep-sea sediments of the South Pacific region demonstrate a progressively increasing trend with depth from $-0.20 \pm 0.02‰$ to $0.44 \pm 0.02‰$[19]. The trends in $\delta^{98}$Mo values in both cores coincide with increasing Mo/Ti ratios (or Mo concentration) and decreasing Mn/Mo and Fe/Mo ratios (Fig. 2 and Supplementary Fig. S2).

## Mo Distribution and Isotopes in Fe-Mn (Hydro)oxide phases
Mo concentrations, their relative proportions in each extraction phase (derived from sequential extractions), and Mo isotope distributions in the Fe-Mn (hydro)oxides phase are summarized in Supplementary Tables S2 and S3. The results indicate that Mo is predominantly associated with the Fe-Mn (hydro)oxides phase. Furthermore, the isotopic

variation patterns of Mo in this phase for both cores closely mirror those observed in the bulk sediment samples (Fig. 2). A detailed evaluation of the sequential extraction methodology and a comprehensive description of the data are available in the Supplementary Information S2.

## Mo in Deep-sea pelagic sediments
Deep-sea pelagic sediments characteristically show enrichment in transition metals, particularly Mo, relative to terrigenous deposits[20–24]. This Mo enrichment, prominently displayed in Indian and Pacific Ocean sediments and evident in cores XT19 and GC112, exhibits strong correlation with Mn and weak correlation with Fe (Fig. 1C, D). The absence of Mo–Ti correlation (Supplementary Fig. S1A) indicates that Mo predominantly associates with Fe-Mn (hydro)oxides, a finding confirmed by sequential extraction experiments showing the majority

of sedimentary Mo bound to this phase (Supplementary Fig. S3A and Supplementary Tables S2 and S3).

Previous research indicates that hydrothermal activity significantly influences Fe and Mn concentrations in South and East Pacific sediments[24,25]. Hydrothermal fluids, introduced into seawater via vent sites or through diffuse flow from flanking regions, transport considerable quantities of Fe and Mn. Subsequent deposition of these metals alters the sediments' geochemical composition. In contrast, the Western Pacific sediments analyzed here contain markedly lower Fe/Ti (<14.0) and Mn/Ti (<6.0) (Fig. 2A, B), suggesting minimal hydrothermal influence. GEOTRACES data further support this interpretation, showing limited hydrothermal impact on dissolved Fe, Mn, and Mo concentrations at sites GPc06 (Central North Pacific) and GP18 (Western Pacific) (Supplementary Fig. S4). Notably, even at GP16 (Eastern Pacific) (Supplementary Fig. S5), where hydrothermal inputs affect oceanic Fe and Mn distributions, Mo concentrations remain largely unaffected. These observations indicate that hydrothermal fluids do not serve as the primary Mo source in these sediments. Rather, consistent with studies of Fe-Mn nodules, crusts, and pelagic sediments[1,18,26], the dominant Mo source appears to be seawater or pore water, likely through dissolved Mo adsorption onto precipitating Fe-Mn (hydro)oxides within the water column or sediment.

## Mo Cycle in deep-sea pelagic sediments

Within both sediment cores, Mn/Ti, Fe/Ti, and Mo/Ti ratios (or Mn and Mo concentrations) increase with depth, while Mn/Mo and Fe/Mo ratios decrease (Fig. 2 and Supplementary Fig. S2). This observed pattern of increasing Mn/Ti and Fe/Ti ratios with depth aligns with previous findings in numerous deep-sea sediment cores[13,15,20–23,27,28]. Two primary mechanisms have been proposed for this Mn and Fe enrichment: (1) reductive dissolution and subsequent re-oxidation, where Mn and Fe (hydro)oxides reduction in pore water generates dissolved Mn and Fe ions that then re-oxidize and accumulate as oxides in deeper sediment layers[27]; and (2) oxidative precipitation, involving direct Fe and Mn precipitation from the water column under oxidizing bottom-water conditions during sediment deposition. The latter mechanism is governed by the availability of Fe and Mn in bottom waters and the sediment accumulation rate.

Bottom waters in the Western Pacific study area exhibit high dissolved oxygen concentrations[16,29]. Pore water profiles from cores in adjacent regions show $SO_4^{2-}$ and $NO_3^-$ concentrations equal to or exceeding those of seawater. Dissolved Fe and Mn concentrations are low (0.05–0.3 μmol/L)[30], contrasting sharply with the significantly higher Fe (25–200 μmol/L) and Mn (16–180 μmol/L) levels typical of pore waters in nearshore reducing environments[31]. These values indicate that predominantly oxic conditions prevail within the sediment column. These conditions align with previous research indicating that modern bottom-water oxygen concentrations in deep-sea environments (typically >4000 m depth) exceed 140 μmol/kg[32], and that oxygen can penetrate to the volcanic basement[29,33,34]. Consequently, enhanced precipitation under oxidizing bottom-water conditions represents the most plausible explanation for the observed Fe and Mn enrichment with depth. Notably, while the dissolved Fe and Mn concentrations in pore water from adjacent regions (0.05–0.3 μmol/L) are low compared to those in reducing environments, they exceed typical seawater levels. Research suggests that, under oxic conditions, such elevated dissolved Fe primarily originates from colloidal Fe released during the non-reductive diagenesis of tephra sediments[35–37]. By analogy, the observed Mn enrichment may also largely result from the release of colloidal Mn via similar non-reductive diagenetic processes involving tephra material.

The Mo/Ti ratios in both sediment cores exhibit variation patterns similar to those of Mn/Ti and Fe/Ti (Fig. 2), suggesting that sedimentary Mo might co-precipitate with Fe-Mn (hydro)oxides. However, if Mo were to deposit and enrich concurrently with Fe-Mn (hydro)oxides,

the Mn/Mo and Fe/Mo ratios in the sediment, along with its Mo isotopic composition, should align with values from the top surface of both sediment cores and remain constant with depth. In contrast, our data show that Mn/Mo and Fe/Mo ratios gradually decrease with depth, while Mo isotopes become progressively heavier. Consequently, we hypothesize that post-sedimentation, sedimentary Mo is likely influenced by early diagenetic processes, such as the infiltration and addition of Mo from bottom waters. This hypothesis aligns with previous findings that early diagenetic processes are primary factors influencing the fractionation of rare earth elements and Zn isotope composition in Pacific Ocean deep-sea sediments[24,30]. This is further corroborated by the observation that dissolved Mo concentrations in the pore waters of Western Pacific sediments are generally lower than bottom water concentrations and decrease with depth (from approximately 110 nM−the seawater Mo concentration−at the sediment-water interface to about 20 nM at 15 meters below the seafloor (mbsf)[38]. This indicates continuous Mo transport from bottom water into the sediment via diffusion and subsequent adsorption onto Fe-Mn (hydro)oxides. Due to higher Mn and Fe content and/or prolonged adsorption in deeper layers, these sediments exhibit greater Mo enrichment than shallower layers. This leads to enhanced Mo accumulation at depth (Fig. 2C and Supplementary Fig. S2A), and Mo/Ti ratios show a positive correlation with Mn/Ti and Fe/Ti ratios (Fig. 3A, B). Conversely, Fe and Mn concentrations in pore water are slightly higher than in bottom water, which precludes their sustained transport from bottom water into sediments. Ultimately, these processes lead to a gradual decrease in the Mn/Mo and Fe/Mo ratios with depth (Fig. 2)[20–22].

## Mechanism of the Mo isotopes in deep-sea pelagic sediments

A key finding in this study is the progressively heavier Mo isotopic signature observed in the deep-sea sediments with increasing depth, contrasting with the lighter signatures found in Fe-Mn nodules and crusts (Fig. 2D). This trend correlates positively with the Mo/Ti ratio and negatively with both Mn/Mo and Fe/Mo ratios (Fig. 3C–E). This variation is likely driven by bottom-water Mo infiltration into the sediment and subsequent cycling within the deeper sediment column. At the sediment-water interface, bottom-water Mo ($\delta^{98}Mo = 2.34 \pm 0.10‰$) undergoes adsorption by Fe-Mn (hydro)oxides in particles or sediments[39–42]. The preferential adsorption of lighter Mo isotopes by these (hydro)oxides (fractionation ~2.9‰) produces a negative Mo isotopic signature at the sediment surface, consistent with observations in Fe-Mn nodules and crusts[1,18,43]. The remaining dissolved Mo, enriched in heavier isotopes, penetrates deeper into the sediment column where it undergoes further adsorption by Fe-Mn (hydro)oxides. Although these (hydro)oxides preferentially adsorb lighter isotopes, the infiltrating pore water's already-heavy isotopic composition results in progressively heavier Mo isotopic signatures in deeper sediments. This hypothesis is supported by research demonstrating that pore water $\delta^{98}Mo$ compositions become progressively heavier with depth due to cyclic adsorption by Fe-Mn (hydro)oxides during early diagenesis, a mechanism that elevates pore water $\delta^{98}Mo$ values to as much as 3.5‰[31,44,45]. Additional factors potentially contributing to heavier isotopic signatures in deeper sediments include progressively reduced fractionation (from ~2.9‰ to 1.0‰) during Fe-Mn (hydro)oxides adsorption, likely caused by variations in the proportion of Fe and Mn (hydro)oxides, or the near-complete capture of Mo from bottom seawater. Notably, maximum $\delta^{98}Mo$ values occur at the bottom of GC112 (5.40–7.35 mbsf), deviating from the core's fitting curve (Fig. 3D). This observation may reflect Fe (hydro)oxides play a discernible role in Mo adsorption at these depths, supported by sequential extraction experiments confirm Mo association with Fe (hydro)oxides (Supplementary Table S2 and Supplementary Fig. S3A) and Fe oxyhydroxides' smaller adsorption-induced fractionation compared to Mn oxide. In summary, the depth-dependent increase in Mo isotopic signatures likely results from a two-stage process: initial

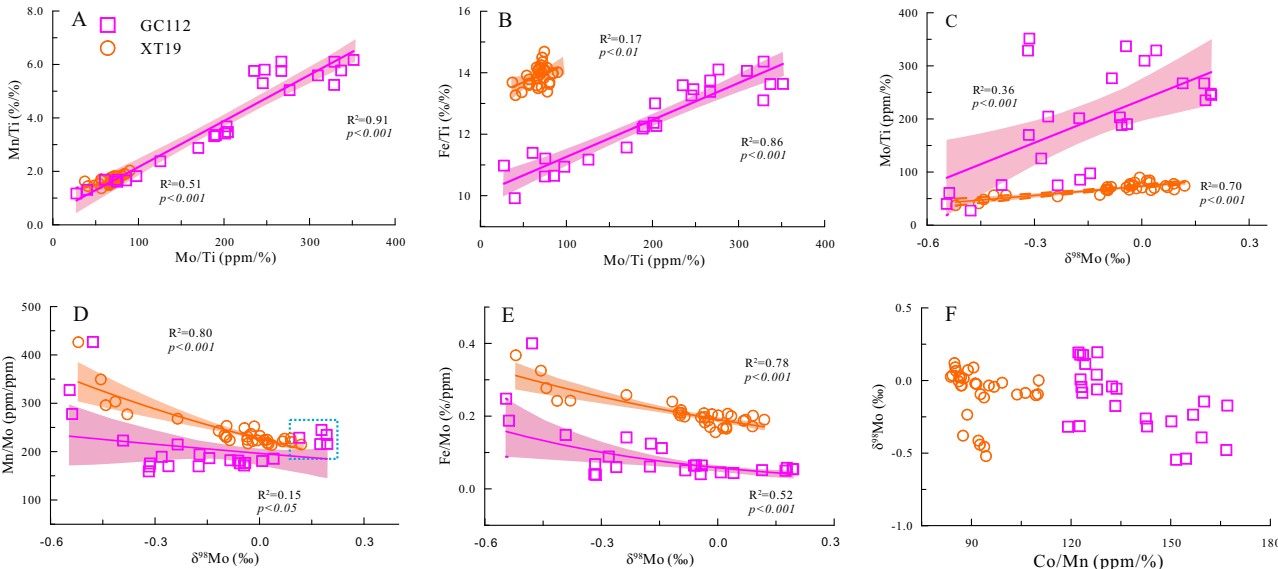

**Fig. 3 | Covariation of Mo isotopes and metal ratios in western Pacific deep-sea sediments.** Cross-plots showing the relationships between: (**A**, **B**) Mo/Ti and Mn/Ti, Fe/Ti; and (**C**–**F**) $\delta^{98}$Mo and Mo/Ti, Mn/Mo, Fe/Mo, and Co/Mn. The shaded areas represent the 95% confidence intervals for the linear regressions.

preferential adsorption of lighter isotopes during bottom water infiltration, followed by adsorption of comparatively heavier isotopes at greater depths, producing a systematic increase in sedimentary Mo isotopic composition with depth.

Chen et al. (2022)[16] attributed exceptionally light Mo isotopic compositions in deep-sea sediments to the preferential release of lighter Mo isotopes from Fe-Mn (hydro)oxides during enhanced dissolution at depths exceeding those they investigated. Indeed, in cores GC112 and XT19 above 4.0 mbsf, our data show a consistent decrease in Mn/Ti, Fe/Ti, and Mo/Ti ratios toward the sediment surface (Fig. 2A–C). However, if Fe-Mn (hydro)oxides dissolution drove this reduction, as Chen et al. (2022)[16] proposed, sedimentary Mo isotopes should become progressively heavier (i.e., exhibit increasing $\delta^{98}$Mo values) upwards from 4.0 mbsf, owing to the preferential release of lighter isotopes. Contrary to this expectation, our data reveal a gradual enrichment in lighter Mo isotopes (decreasing $\delta^{98}$Mo values) upwards from this depth (Fig. 2D). Therefore, this proposed mechanism does not adequately explain the Mo isotopic variations observed in our study.

Similarly, Ahmad et al. (2021)[19] attributed the observed increase in Mo isotopic values (signifying an enrichment in heavier isotopes) with depth in deep-sea sediments to the preferential release of light Mo isotopes during the reductive dissolution of Fe-Mn (hydro)oxides. They posited that with increasing sediment depth, this process preferentially releases lighter Mo isotopes, leading to a progressive decrease in solid-phase Mo content and a concomitant enrichment of heavier isotopes in the residual sediment. Consequently, a strong negative correlation between Mo concentration and its isotopic composition would be anticipated. In contrast, our core samples exhibit contrary trends: Mo/Ti ratios (or Mo concentrations) increase with depth (Fig. 2C and Supplementary Fig. S2A), accompanied by a positive correlation between Mo/Ti ratios and isotopic composition (Fig. 3C). Therefore, the mechanism proposed by Ahmad et al. (2021)[19] also fails to adequately explain the Mo concentration and isotopic signatures observed in our study.

Sedimentation rate emerges as a key factor influencing metal stable isotopic composition in marine deposits. Recent studies by Fleischmann et al. (2023) and Zhang et al. (2024) demonstrate that, in the absence of post-depositional redox-driven diagenetic alterations, accumulation rates and incorporation mechanisms account for the lighter Ni and Zn isotopic signatures in deep-sea sediments relative to

Fe-Mn crusts[24,26]. The Co/Mn ratio, derived from ferromanganese nodules unaffected by early diagenesis, serves as a proxy for this geochemical accumulation rate. Our research reveals a significant difference in the Co/Mn ratio between the XT19 and GC112 cores (Fig. 3F). If the sedimentation rate were to substantially impact the Mo isotope composition of deep-sea sediments, a corresponding discrepancy would be expected in the Mo isotope records of these cores. However, the Mo isotope profiles exhibit a relatively consistent pattern across both locations (Fig. 3F). Furthermore, the correlation between $\delta^{98}$Mo values and the Co/Mn ratio is not statistically significant in the sediments of either core (Fig. 3F). This suggests that the sedimentation rate has a minimal influence on Mo isotope variations in Pacific Ocean deep-sea sediments. Recognizing that the geochemical behavior of Co in deep-sea sediments may be modified by early diagenesis, which would prevent the Co/Mn ratio from accurately recording the sedimentation rate, we utilized the paleomagnetic age model-based sedimentation rate from the XT19 core to further investigate this relationship. The results indicate that sedimentation rates in the XT19 core were relatively stable, measuring approximately 175 cm myr$^{-1}$ in the upper section (< 2.0 mbsf) and 202 cm myr$^{-1}$ in the lower section (>2.0 mbsf)[46]. Despite this rate stability, notable variations were observed in the corresponding Mo isotopic compositions between these two intervals. This finding reinforces the conclusion that sedimentation rate exerts minimal influence on Mo isotope variations in these Pacific Ocean deep-sea sediments.

**Implication for Oceanic Mo isotope, mass balance**

A comprehensive understanding of the oceanic Mo isotopic budget is fundamental for utilizing Mo isotopes as paleoenvironmental proxies[8,12]. While Fe-Mn crusts and nodules have traditionally represented oxic sedimentary Mo isotopes[1,8,12], their slow accumulation rates and relatively limited areal and volumetric distribution on the global ocean floor minimize their contribution to oceanic Mo removal[12]. The dominant Mo removal mechanism likely involves Fe-Mn (hydro)oxides microparticles within pelagic and hemi-pelagic sediments[1,12–14], positioning deep-sea sediments as critical end-member sinks in isotopic studies[12]. However, existing Mo isotope data for deep-sea sediments remains limited[15,16]. This study addresses this knowledge gap by analyzing deeper marine sediments and integrating results with existing Mo content data, providing robust

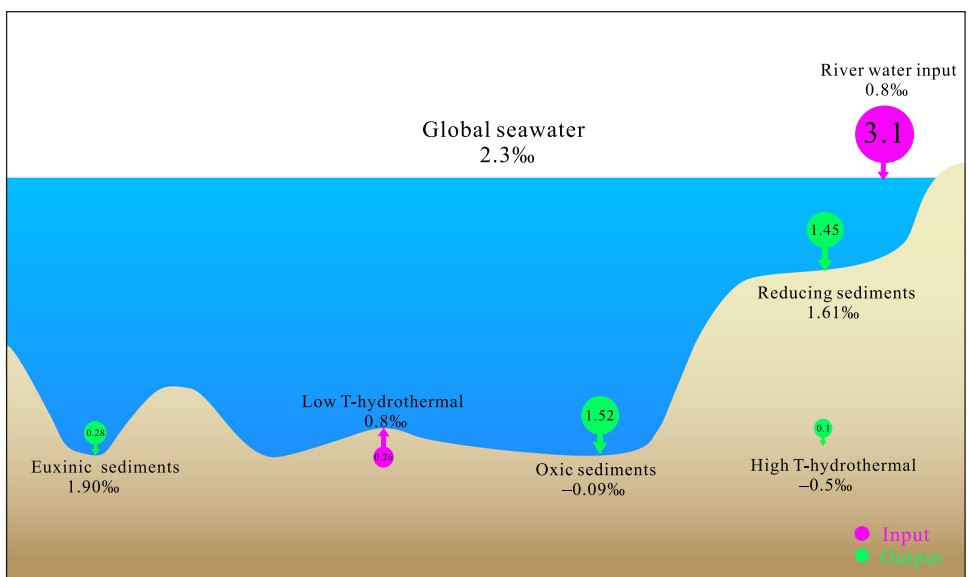

**Fig. 4 | An updated model of the oceanic molybdenum (Mo) isotopic mass balance.** This schematic illustrates the primary inputs (purple circles: riverine, low-temperature hydrothermal) and outputs (green circles: various sediments, high-temperature hydrothermal) for the global Mo budget. The size of each circle is proportional to the estimated flux with values noted in black ($\times 10^8$ mol yr$^{-1}$). The model is conceptually based on the framework of Little et al. (2025)[12].

constraints on Mo output flux estimations and isotopic composition from oxidized sediments.

Deep-sea pelagic sediments significantly influence the oceanic Mo cycle. We estimated their Mo flux using three parameters: authigenic Mo content, mass accumulation rate, and areal extent. Pelagic clay, covering 40.7% of the ocean floor[47], accumulates at ~2.2 g m$^{-2}$ yr$^{-1}$[14,48]. Combining our data with Mo concentration data from Indian, Atlantic, and Pacific Ocean sediments (over 2000 Mo concentration data)[13,15,16,20–22] yields an average authigenic Mo concentration of $45.0 \pm 30$ ppm (a 0.5 m high-resolution depth-averaging method, details of calculation method in Supplementary Information S3). This produces a calculated Mo flux of $1.52 \times 10^8$ mol yr$^{-1}$, substantially exceeding previous estimates of $0.09$–$0.35 \times 10^8$ mol yr$^{-1}$ (based on authigenic Mn accumulation rates of 2.3–6 µmol cm$^{-2}$ yr$^{-1}$ and Mo/Mn ratio of 0.018) and $0.87 \times 10^8$ mol yr$^{-1}$[12,49]. This disparity likely arises from previous studies neglecting Mo adsorption by Fe (hydro)oxides or underestimating sedimentary Mo concentrations. Sequential leaching experiments confirm this observation, indicating that a substantial proportion of the total Mo content is associated with the Fe (hydro)oxides phase (Supplementary Table S2 and Supplementary Fig. S3A).

The average authigenic $\delta^{98}$Mo value of deep-sea sediment, calculated from previous studies[15,16] and our investigation, yields $-0.09 \pm 0.23$‰ (a 0.5 m high-resolution depth-averaging method, details of calculation method in Supplementary Information S3). Deep-sea sediments, as integral components of the modern ocean system, play a vital role in regulating marine geochemical cycles through early diagenetic processes. Previous research indicates that seawater infiltration rates—a key driver of early diagenesis in deep-sea settings—are relatively high, ranging from 1.2 to 10 cm year$^{-1}$[50,51]. Based on the core depths analyzed in this study, the estimated timescale for seawater infiltration spans several decades to several hundred years. This duration is substantially shorter than the oceanic Mo residence time of 0.44 Ma[12]. Furthermore, the global oceanic Mo isotopic composition, reconstructed from Fe-Mn nodules since the Cenozoic era, has remained stable[1]. Consistent with this stability, Mo isotope compositions in the studied cores have undergone only minor changes since their deposition approximately 10 million years ago (Supplementary

Fig. S5). Collectively, these observations suggest that the exchange of Mo isotopes between deep-sea sediments and the ocean has largely been in a stable equilibrium state since the time of sediment deposition. Therefore, the average $\delta^{98}$Mo value ($-0.09 \pm 0.23$‰) derived from these deep-sea sediments provides a reliable proxy for the oxic sink value in the modern ocean.

## A mass balance for oceanic Mo and its isotopes

Current estimates of global oceanic Mo output flux and isotopic composition in sediments lack precision, particularly regarding variations in reducing and euxinic settings (Supplementary Table S4). Through integration of our results with previously reported fluxes and $\delta^{98}$Mo values for reducing and euxinic environments (Supplementary Table S4), we developed enhanced constraints on oceanic Mo output based on the mass balance equations:

$$F_{RIV} + F_{L-hyd} = F_{OX} + F_{RED} + F_{EUX} + F_{H-hyd} \tag{1}$$

$$\begin{aligned} F_{RIV} \times \delta^{98}Mo_{RIV} + F_{L-hyd} \times \delta^{98}Mo_{L-hyd} \\ = F_{OX} \times \delta^{98}Mo_{OX} + F_{RED} \times \delta^{98}Mo_{RED} + F_{EUX} \times \delta^{98}Mo_{EUX} \\ + F_{H-hyd} \times \delta^{98}Mo_{H-hyd} \end{aligned} \tag{2}$$

Where F terms represent Mo fluxes ($10^8$ mol/yr) from various environments: Riverine ($F_{RIV}$), Low-temperature hydrothermal ($F_{L-hyd}$), High-temperature hydrothermal ($F_{H-hyd}$), Oxic ($F_{OX}$), Reducing ($F_{RED}$), and Euxinic ($F_{EUX}$) sedimentary environments. The corresponding $\delta^{98}$Mo terms denote Mo isotope compositions for each source or sink, with values and ranges presented in Supplementary Table S4.

Monte Carlo analysis determined optimal flux and isotopic signatures for equilibrium between terrestrial input and marine sinks (details of calculation method in Supplementary Information S4). The calculations indicate a Mo output flux to reducing conditions of $1.45 \times 10^8$ mol yr$^{-1}$ ($\delta^{98}$Mo = 1.61‰) and to euxinic conditions of $0.28 \times 10^8$ mol yr$^{-1}$ ($\delta^{98}$Mo = 1.90‰). When combined with the oxic sink flux of $1.52 \times 10^8$ mol yr$^{-1}$ ($\delta^{98}$Mo = $-0.09$‰), these fluxes align with the reported terrestrial Mo input of $3.36 \times 10^8$ mol yr$^{-1}$ ($\delta^{98}$Mo = 0.80‰), suggesting a balanced oceanic Mo cycle that

incorporates the newly quantified oxic Mo sink associated with pelagic sediments (Fig. 4). The uncertainties of the calculated values were evaluated by the Monte Carlo simulations (Supplementary Fig. S7). Analysis reveals that variations in $F_{RED}$ and $\delta^{98}Mo_{RED}$ substantially influence the total error in the equilibrium calculation, which is determined as the absolute sum of errors from both the Mo mass balance and isotopic balance. In contrast, $F_{EUX}$ and $\delta^{98}Mo_{EUX}$ exert minimal influence.

These results strongly support the hypothesis that pelagic deep-sea sediments represent a significant isotopically heavy Mo sink compared to Fe-Mn crusts and nodules. Furthermore, the findings emphasize the critical role of deep-sea sediments as a more substantial oxic Mo sink than Fe-Mn crusts and nodules, underscoring the necessity of including them in global Mo cycle mass balance models.

## Implications for ancient Oceanic Mo isotope composition

Our findings enhance the understanding of the modern oceanic Mo cycle. Specifically, we have refined the output flux of Mo to oxic sinks, now estimated at $1.52 \times 10^8$ mol yr$^{-1}$. This value constitutes 45% of total outputs, surpassing the contributions of other sinks (Supplementary Table S4). Additionally, we propose a heavier isotopic composition ($\delta^{98}Mo = -0.09 \pm 0.23$‰) for the oxidized Mo end-member. These updated parameters necessitate a re-evaluation of previously published records of Mo content and isotopes, which are essential for accurately estimating the global expansion and reduction of oceanic euxinia throughout Earth's history[6]. To evaluate the impact of these revisions, the Mo mass balance model (Supplementary Materials S5 for a detailed model description and parameters) was employed[6,9]. This model assessed how changes in the Mo isotopic composition of the oxic sink end-member affect reconstructed estimates of global euxinic seafloor area, as inferred from ancient ocean Mo isotope composition. The evaluations focused on five distinct ancient ocean Mo isotope compositions: three from key geological intervals—the late Paleoproterozoic (1.2‰)[8], the Early Jurassic (1.6‰)[52], and the Paleocene-Eocene Thermal Maximum (PETM) (2.0‰)[53]—and two hypothetical values (1.4‰ and 1.8‰).

The results demonstrate substantial alterations in the calculated relative extents of global euxinic seafloor across these intervals when the oxic sink end-member is modified—specifically, when the fractionation factor ($\Delta^{98}Mo_{seawater-oxic\ sink}$) shifts from 3.0‰ to 2.40‰ (Fig. 5). Assuming the relative area of oxic seafloor [$F_{OX}/(F_{OX} + F_{RED})$] is held constant in Fig. 5, the reconstructed relative extent of global euxinic seafloor is lower than previously indicated[8,52,53]. In other words, prior estimates appear to have significantly overestimated the relative extent of euxinic seafloor.

Our study may also promote a deeper in understanding the forms of Mo occurrence in oxic sedimentary environments. While Mn oxides have traditionally been regarded as the primary form of Mo in marine oxidizing sediments, our research demonstrates that Fe (hydro)oxides may constitute a subordinate reservoir, accounting for a certain proportion of Mo occurrences (Supplementary Fig. S3A). In essence, Mn and Fe (hydro)oxides collectively regulate Mo enrichment in deep-sea sediments. This conclusion is further substantiated by the strong correlations observed between Mo/Ti ratios and both Mn/Ti and Fe/Ti ratios in bulk sediment samples (Fig. 3), relationships that persist across different sequential extraction phases (Supplementary Fig. S8). Given the substantial differences in Mo adsorption fractionation between Mn and Fe (hydro)oxides, we propose that the cycling of these (hydro)oxides throughout geological history is closely linked to variations in oceanic Mo isotope composition, as they influence the isotopic signature of the oxic sink end-member. This discovery may provide an explanation for the dramatic fluctuations in oceanic Mo isotope composition observed throughout Earth's history and advances our understanding of the complex processes driving the chemical evolution of Earth's oceans[6,7].

## Methods

### Sediment cores sampling

Deep-sea pelagic sediments were collected from two gravity piston cores in the western Pacific Ocean: XT19 (15°42′00″N, 133°28′48″E) from the Central Philippine Sea Basin and GC112 (16°54′06″N, 162°10′47″E) from the Pigafetta Basin[24,30,46,54] (Fig. 1). The XT19 core was retrieved in 2019 by the Qingdao Institute of Marine Geology, China Geological Survey, at a water depth of 5631 m. Sediment samples from this core, ranging from 0 to 4.5 m below the seafloor (mbsf), consist of light-brown to brown, homogenous siliceous clay and carbonate-free mud[46]. The GC112 core was collected in 2018 by the Guangzhou Marine Geological Survey aboard the research vessel "Haiyang-6" at a water depth of 5777 m. This core, sampled from 0 to 7.5 mbsf, is characterized by a uniform, granular texture and primarily consists of siliceous clay from 0 to 2.25 mbsf, transitioning to zeolitic pelagic clay from 2.4 to 7.5 mbsf[54].

### Major and trace elements

Elemental analyses of both GC112 and XT19 sediments followed established protocols. For GC112, major and trace element concentrations were determined via X-ray fluorescence (XRF) and inductively coupled plasma mass spectrometry (ICP-MS) at the Guangzhou Marine Geological Survey (GMGS), achieving < 5% relative standard deviation (RSD) for replicate analyses[24,54]. XT19 sediment samples underwent HF–HCl–HNO$_3$ digestion before trace element analysis via ICP-MS (Thermo Icap Qc; <5% RSD) and major element analysis via ICP-AES (Thermo Element II; <3% RSD) at the State Key Laboratory of Isotope Geochemistry (Guangzhou Institute of Geochemistry)[55,56]. Complete methodological details are available in Zhang et al. (2024) and Bai et al. (2025)[24,54]. Results are presented in Supplementary Table S1.

### Mo isotopes measurement

Molybdenum isotope compositions were determined using a double-spike method[57]. Briefly, 5–30 mg of sediment sample was precisely weighed into a 15 mL PFA beaker and mixed with a $^{100}Mo-^{97}Mo$ double-spike solution before digestion overnight at 120 °C with about 6 mL of a 2:1 HF (22 mol L$^{-1}$) + HNO$_3$ (14 mol L$^{-1}$) mixture. After drying at 120 °C, the residue was dissolved in 1 mL conc. HCl and again evaporated to dryness. The residue was finally dissolved in 2 mL of a mixture of HF (0.1 mol L$^{-1}$) + HCl (1 mol L$^{-1}$) for chromatographic separation.

Mo was separated and purified following Li et al. (2014), using a custom-made chromatographic N–benzoyl–N–phenyl hydroxylamine resin[57]. Mo isotope ratios were determined by multicollector (MC)–ICP–MS (Thermo-Fisher Scientific Neptune Plus). Isotopic compositions of Mo are expressed as $\delta^{98}Mo$ values relative to the US National Institute of Standards and Technology (NIST) standard reference material (SRM) 3134. Calibration procedures followed Siebert et al. (2001), Rudge et al. (2009), and Zhang et al. (2015)[43,58,59]. Here, all data, including those published previously, are reported relative to NIST SRM 3134 + 0.25‰[60]. For quality-assurance purposes, a NIST SRM 3134 solution and three standard materials (IAPSO seawater, BHVO-2, and W-2a) were repeatedly analyzed together with the samples with results as follows (± 2 SD): NIST SRM 3134, 0.25‰ ± 0.07‰ ($n = 35$); IAPSO seawater, 2.31‰ ± 0.06‰ ($n = 3$); BHVO-2, 0.26‰ ± 0.05‰ ($n = 5$); and W-2a, + 0.15‰ ± 0.07‰ ($n = 5$). These results are consistent with those reported previously[61–63], within analytical error. Procedural Mo blanks for bulk samples contained < 0.5 ng Mo, much less than the total Mo in samples. The Mo isotope results are presented in Supplementary Tables S1 and S3.

### Sequential extraction procedure

A six-step sequential extraction was employed on GC112 sediment[54,63–65] to investigate Mo distribution within Fe-Mn (hydro)oxides. This procedure targeted the following phases: (1) phosphate, (2) easily

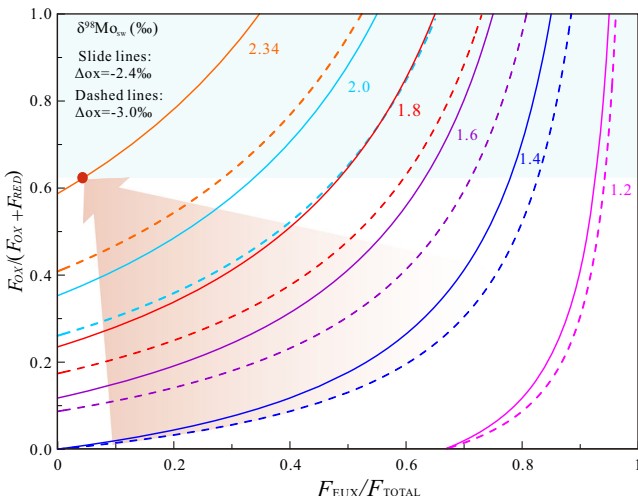

**Fig. 5 | Modeling seawater δ⁹⁸Mo as a function of oceanic sink distributions.** The model illustrates the steady-state seawater $\delta^{98}$Mo value based on the relative proportions of euxinic ($F_{EUX}$), reducing ($F_{RED}$), and oxic ($F_{OX}$) sinks. The red dot indicates the modern Mo isotope budget. The light blue-shaded area represents unrealistic mass balance solutions. The shaded region, highlighted by an arrow, illustrates the projected trend in seawater $\delta^{98}$Mo values with increased deep-ocean oxygenation (Modified from Chen et al., 2015)[6].

reducible Mn oxides, (3) easily reducible Fe (hydro)oxides, (4) moderately reducible Fe (hydro)oxides, (5) crystalline Fe (hydro)oxides, and (6) residual. Briefly, 0.1 g of bulk sediment powder underwent six 4 h extractions using 20 mL of 1 M acetic acid/sodium acetate buffer (pH 4.5) to isolate the exchangeable phase. Easily reducible Mn oxides were extracted with 20 mL of 0.02 M hydroxylamine hydrochloride (pH 2) for 24 h. Subsequent extractions targeted Fe (hydro)oxides: easily reducible (0.1 M hydroxylamine hydrochloride, pH 2, 2 h), moderately reducible (1 M, pH 2, 48 h), and crystalline (1 M, pH 2, 85 °C, 6 h). The final step isolated the aluminosilicate residual phase. Between each step, samples were centrifuged (4000 rpm, 10 min), supernatants filtered (0.2 μm polyethersulfone membrane), and residues washed with Milli-Q H₂O for 2 h. The final residue was washed, dried, and digested using 4 mL concentrated HNO₃/HF (1:3). Elemental analysis (major and trace) of each phase was performed via ICP-AES/ICP-MS. For the Mo distribution analysis, step two isolated the Mn oxide phase, while steps three through five comprised the Fe (hydro)oxides phase. Results are presented in Supplementary Table S2 and Supplementary Fig. S3. It is important to note that this method has inherent limitations. A substantial portion of Fe and Mo remained in the residual fraction suggesting that the extraction of crystalline Fe (hydro)oxides in step 5 may have been incomplete. Furthermore, significant uncertainty exists in quantifying Mo within specific phases due to the potential for partial dissolution and elemental redistribution during the extraction process, as the reagents are operationally defined and not perfectly selective for individual mineral hosts.

To analyze the Mo isotope composition specifically within the Fe-Mn (hydro)oxides fraction, a second sequential extraction targeting both phosphate and Fe-Mn (hydro)oxides was performed on samples from cores GC112 and XT19. The methodology followed established protocols: samples were initially treated with a 1 M acetic acid/sodium acetate buffer (pH 4.5) for 24 h, followed by treatment with a mixed solution of 1 M hydroxylamine hydrochloride and 1 M HCl at 85 °C for 6 h. Major and trace element analyses were conducted on both extracted fractions, while isotopic measurements were performed exclusively on the Fe-Mn (hydro)oxides phase. The elemental composition of the residual fraction has not yet been characterized.

## Data availability

The supplementary data employed are accessible in the "Supplementary Dataset 1, 2" file on Mendeley Datasets[66].

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

## Acknowledgements

We thank Lian Peng, Xiangyu Li, Jianghao Bai, and Hao Wu of the State Key Laboratory of Deep Earth Processes and Resources, GIG-CAS, for their assistance with Mo isotopic and elemental measurements and sequential sediment extraction. We thank Prof. Yinan Deng of the Guangzhou Marine Geological Survey for providing the samples. We thank Prof. Mang Lin, Xinming Chen, Lu Yin, Wenfeng Deng, and Yinan Deng for helpful comments and suggestions. This research was financially supported by the National Key Research and Development Program of China (2024YFF0807803 to G.J.W. and 2022YFF0800501 to J.L.M.), the National Natural Science Foundation of China (42473004 to Z.B.W.), the Project of the China Geological Survey, Ministry of Natural Resources (DD20230647 to B.Q.H.), and Guangdong Basic and Applied Basic Research Foundation (2025A1515011045 to Z.B.W.). This is contribution No. IS-3718 from GIGCAS.

## Author contributions

G.J.W. and Z.B.W. conceived and designed the study. Z.B.W., J.L and L. Zhang carried out Mo isotopic measurements. Z.B.W., L. Zou., G.J.W., J.L. and J.L.M. performed the data analyses. B.Q.H. and X.D. organized the collection of XT19 sediment samples. Z.B.W., L. Zou., G.J.W., J.L. and J.L.M. wrote the initial draft of the paper, and all authors discussed the results and contributed to the final manuscript.

## Competing interests

The authors declare no competing interests.
