## [Transparent Peer Review file · Nature Communications]

Revised Oceanic Molybdenum Isotope Budget from Deep-Sea Pelagic Sediments

Corresponding Author: Dr Zhibing Wang

Version 0:

Reviewer comments:

Reviewer #1

(Remarks to the Author)

This is an important contribution to understanding the Mo and Mo isotope cycle that I think should definitely be published after minor to moderate revisions. The findings are provocative and have potentially huge implications for reinterpreting past paleoceanographic studies. The paper is generally well written and I see no issues with the data or interpretation although I really wish the authors would more fully take the plunge at the end and elaborate more on the implications for using Mo isotopes as a paleoceanographic redox tool. The authors seem to walk up to the edge of that ledge but I think the paper would be better if they took the plunge. Below are some line by line comments and suggestions.

The language or word choice in the first sentence is not great. Specifically, the phrase “dual role” doesn’t fit particularly well because you are talking about two totally unrelated things (i.e., “apples and oranges”). The first role is a role in biological processes whereas the second is in a completely unrelated topic: scientific inquiry.

Lines 80-82. It might be worth providing an additional sentence here to remark in more detail on WHAT the implications are.

-
Please indicate how you are reporting your $d_{98}\text{Mo}$ values early on in the main text of the paper. In other words, are you setting NIST3134 to 0 per mil (Goldberg) or adding a “+0.25” per Nagler.

Lines 85-86. To me, it looks like the Mo/Al and especially Mn/Al of site XT19 do not really change with depth. Also, in the sentence beginning in line 85, it makes more sense to say “Mn/Al ratios and Mo/Al...” since this is the order that you show them in the data plots.

Line 88 – Either get rid of the semicolon or get rid of the “whereas”

Lines 93-95. It’s quite striking how similar the highest and lowest values are in these two very distant localities!

Line 96 – “patter” should be pattern

Line 97 – “ang” should be and

Line 98 – What do you mean by “bottom layer”?

Line 111 – When you initially mention hydrothermal activity, I thought you meant general input of metals to oceanwater from hydrothermal vents but it sounds like what you are actually talking about is in situ deposition of metals within the sediment by hydrothermal seeps from directly below? Maybe clarify this.

Section running from about Line 140 to 153 – If the Fe-Mn oxides higher in the sediment section are being reductively dissolved, does this say something about the redox gradient in the sediment package that might affect Mo solubility given that its behavior is quite different than Fe and Mn? In other words, would this inhibit Mo movement deeper into the sediment

pile? Or do you need outright sulfidic conditions for this to be an issue for Mo?

Line 208-210 – The crusts form slowly but how do the Mo concentrations in these compare to the sediments?

Paragraph starting at line 285 – It would be great if you could elaborate on the implications for previous interpretations of Mo isotope data in ancient sediments! Just one example: what bearing does your finding have on debates over the redox state of the Proterozoic oceans?

Figure 1. The data plots really need larger text type. Also, why not have Mo on the y-axis in both of the data plots?

Line 548-550 – There are no light purple bars in this figure. Also, the last two sentences in the figure caption seem to be saying precisely the same thing; maybe you forgot to delete one of them before submitting this?

Figure S3. So do the A, B, C, and D within each plot (marked on the horizontal plane) stand for phosphate, Mn oxide, Fe h-oxide, and residual? This should be spelled out in the caption.

Figure S4. These would be interesting diagrams, but the text components are small to the point of being unreadable.

Reviewer #2

(Remarks to the Author)

Review of the manuscript NCOMMS-25-10252 “Revised Oceanic Mo Isotope Budget from Deep-Sea Pelagic Sediments” by Wang et al. submitted to Nature Communications.

This manuscript presents stable Mo isotope composition from two Pacific Ocean deep-sea pelagic sediment cores. Cores do show trend with increasing Mo isotope ratios with depth and covariation with Mo/Al with Mo/Al and Mn/Mo ratios. The authors argue that the depth-dependent increase in Mo isotope signatures are likely results of an initial preferential adsorption of lighter Mo isotopes during bottom water infiltration coupled by adsorption of heavier Mo isotopes at greater depths. Based on the presented data I think this is a possible scenario and explanation for the increasing Mo isotope signatures with increasing core depth. I also have no doubt that the data presented here are of good quality. However, I have some small but also some more substantial comments concerning the influence of Mn diagenesis on the Mo isotopes composition, the use of Co as tracer geochemical accumulation rate and influence of Fe-oxides on Mo in these sediments. These require revision.

Line 85: “...in both sediment cores. Mo/Al ratios and Mn/Al ratios (or concentrations) increase with...”

Please specify the “or concentration”. Ratios or concentrations?

Line 87: “in XT19, they show a continuous increase with depth;”

Considering Fig.2F it is not a continuous increase. There are quite some fluctuations the first 2 m down-core

Line 87: “Core GC112 demonstrates a monotonic increase in $\delta^{98}\text{Mo}$ values with depth...”

I agree with the authors that there is an increase in $\delta^{98}\text{Mo}$ values with depth, but it is not monotonic.

Line 97: “...0.03‰ at 3.0 mbsf, and then decrease to 0.50‰ at the bottom layer”

Please correct “and”

Line 98: “the citation number 19”

Indeed, as the authors state the Goldschmidt abstract they cite shows an increase in $\delta^{98}\text{Mo}$ with burial depth. However these authors state that the positive relation between Mo content and isotope composition indicates that the Mo carrier (i.e., Fe-Mn (oxyhydr)oxides) might preferentially release the isotopically heavy Mo during Mn reduction.

On the other hand, another published article where some authors of this submitted manuscript are listed as co-authors “Extremely light molybdenum isotope signature of sediments in the Mariana Trench” published in Chemical Geology from Chen et al 2022. There extremely light $\delta^{98}\text{Mo}$ down to -1.71 ± 0.05 ‰. Here light $\delta^{98}\text{Mo}$ values of sediments in deeper parts of the cores coincide with enhanced Mo accumulation. Here the authors state that the source of the isotopically light dissolved Mo was explained by the preferential release of isotopically lighter Mo from Fe-Mn (oxyhydr)oxides during the enhanced dissolution at deeper depth.

So both, the cited abstract and Chen et al 2022 state that Fe-Mn (oxyhydr)oxides diagenesis/reduction has an influence on the sediment $\delta^{98}\text{Mo}$ pattern along a core profile.

Line 128: “Previous studies have documented increasing Mn concentrations with depth in pelagic sediments, attributed to progressive Mn oxidation in porewaters leading to enhanced Mn oxide accumulation in deeper sediment layers.”

For citation 28 from Kato et al 2011 “Deep-sea mud in the Pacific Ocean as a potential resource for rare - earth elements”, I was not able to find such a statement for increasing Mn concentrations with depth.

Line 154: “A key finding in this study is the progressively heavier Mo isotopic signature observed in the deep-sea sediments with increasing depth, contrasting with the lighter signatures found in Fe-Mn nodules and crusts (Fig. 2C).”

I think it is fair to mention that this finding has also been discovered by Ahmad et al 2021 for a Mn- rich Pacific Ocean drill core “Decreasing Mo/Mn and higher $\delta^{98}/^{95}\text{Mo}$ of sediments with core depth and depositional age potentially record an additional early diagenetic process” However this has been interpreted to “resulting in Mo loss and fractionation likely related to the aging of Mn-oxyhydroxides during sediment deposition and compaction, which causes a higher structural order and lower reactivity of the Mn-phases.”

This is different to the interpretation presented here.

Line 189: “Our research reveals a negative correlation between $\delta^{98}\text{Mo}$ values and the Co/Mn ratio (a proxy for geochemical accumulation rate) in deep-sea sediments (Fig. 5),...”

A reference is needed for Co/Mn as proxy for geochemical accumulation rate. For me this is not conclusive: Co concentration

correlate with Mn and Mo concentrations ($R=0.89$ between Mo and Co). I somehow expect these since Co and Mo both are enriched in Mn-oxides. However, as Co shows the same concentration pattern as Mn and Mo and both are affected by diagenesis and/or migration through pore-water, how Co/Mn ratios can be a proxy for geochemical accumulation rate.

Line 209: "However, existing Mo isotope data for deep sea sediments remains limited, with previous Pacific Ocean studies focusing solely on surface layers (<30 cm)"

Not quite right, see Ahmad et al 2021

Line 237 - 246 "Our samples, representing only ~7.5 m of deep-sea sediments, likely.... Consequently, the mean $\delta^{98}\text{Mo}$ of the entire deep-sea sediment column is likely higher than $-0.09 \pm 0.23\text{‰}$ and significantly exceeds the $\delta^{98}\text{Mo}$ observed in Fe and Mn oxide crusts and nodules ($-0.70 \pm 0.10\text{‰}$).

This observation is supported by Ahmad et al 2021 also for sediment below 7.5 m and should be cited here.

Does this 2% represent the fraction of pelagic clays relative to the GLOSS? Is this true why they represent the uppermost 2%? Deep-sea pelagic sediments make up only a small portion of the global subducting sediment GLOSS (See Plank, The Chemical Composition of Subducting Sediments, 2014), with Sediment thickness for dominant Pelagic clay lithologies reaching ~70m (see Tonga).

Line 268: "Analysis reveals that variations in FRED and $\delta^{98}\text{MoRED}$ substantially influence the total error in the equilibrium calculation, calculated as the absolute sum of the Mo mass balance and isotopic balance errors. In contrast, FEUX and $\delta^{98}\text{MoEUX}$ exert minimal influence."

Fig S6, where the Parameter Distributions are shown the y-axis displays Mo isotope ratios in ‰ and the range of fluxes in 108 mol yr^{-1} – this should be better indicated.

Another question regarding the position of the optimal solution (star) in Fig S6. For me the total Error is minimal for a large range of $\delta^{98}\text{MoEUX}$ and FEUX. So how the optimal solution (star) in Fig S6 has been constrained?

Line 286: "While Mn oxides have traditionally been regarded as the primary form of Mo in marine oxidizing sediments, our research demonstrates that Fe oxides represent another critical form, accounting for over half of Mo occurrences (Fig. S3)" This comes a little bit of surprise for me. Not that I do not agree that Fe-oxides can have an influence especially in setting with abundant Fe-oxide formation. However, calculating the authigenic Fe and Mn fraction within the sediment using the major element UCC concentrations from Rudnick & Gao as detrital input, as it was used for Mo by the authors, e.g. 80-90% of the Mn is authigenic compared to only 20-30% of Fe. Accounting the higher. A high correlation coefficient of 0.92 and 0.65 and between authigenic Mo and Mn concentration for core GC112 and TX12 respectively, indic

Reviewer #3

(Remarks to the Author)

Wang et al. presented new Mo isotope data from two cores of deep-sea pelagic sediments, aiming to revise the Mo isotopic fractionations between seawater and its oxic sink, and reconstruct the global marine Mo budgets. This work is very important for the community to understand global marine Mo isotope cycle and use Mo isotope as a paleo-redox proxy, as the pioneer research did not fully solve this issue by only investigating Fe-Mn crust. The manuscript is also well organized and written. However, several issues should be addressed before it can be published. Please see my comments below.

Major concerns:

1. The Mo isotope signals in bulk marine sediments were reported in the current study, although the sequential extraction of Mo in different phases were also conducted. It is quite important to clarify the Mo phases in bulk sediments, thus it will be helpful if the authors can analyze Mo isotopes in Mn-Fe oxides, besides the bulk sediments. Additionally, the current manuscript didn't discuss the results of sequential extraction of Mo, even though there are only elemental concentration data. I recommend the authors further describe and discuss this part in the revised version.

2. Changes in scavenging patterns of Mo within the sediment piles. The key finding in this study is the increased $\delta^{98}\text{Mo}$ values with water depth, positive correlations between $\delta^{98}\text{Mo}$ and Mo/Al, and negative correlations between $\delta^{98}\text{Mo}$ and Mn/Mo ratios. However, this observation notably supports that the Mo precipitation in the deeper sediment pile may reflect the authigenic Mn accumulation under more reducing conditions. Additionally, some reducing sediments can also have high Mn contents (e.g., Gullmar Fjord, Sweden) (Goldberg et al., 2012 Chemical Geology). Thus, one critical point is to identify the types of Mn burial and redox conditions of pore water within the sediments, which should be fully constrained in this study. Any pore water data or redox-sensitive elemental analysis (e.g., Mo-U-V) should help to clarify this point.

3. The effects of isotopic fractionations from deeper sediments on modern global seawater Mo mass balance. The residence time of Mo in modern ocean is around 400 kyr. If the deeper part of the sediment profile in this study was deposited much earlier than 400 ka, the variations of their Mo isotope compositions may not influence modern seawater Mo budgets. It will be helpful to present the depositional age or relative age model of the studied sediment profiles.

Specific items:

Line 34. Better to say "proxy for ancient oceanic oxygen levels".

Line 58. The reference 7 (Gill et al., 2011 Nature) didn't mention Mo proxy. It's better to cite Wei et al. (2021) Earth-Science Reviews, instead.

Line 112-114. Why can low Fe and Mn concentrations of the sediments denote limited hydrothermal influence? I think the dilution of detrital materials (Si, Al) should be considered. Thus, the ratios of Fe/Al, Mn/Al can be used to trace the source of oxides in these sediments.

Line 130-131. Progressive Mn oxidation in pore waters should lead to the decreases in dissolved Mn through the sediment pile. In this view, the deeper sediment layers should have lower Mn accumulations. Does high Mn concentration of deeper sediments represent the accumulation of Mn oxides, or other Mn-bearing minerals (rhodochrosite)?

Line 148-150. Do lower Mn/Mo and Fe/Mo ratios of deeper sediments reflect the effects of local pore water redox condition

on Mo scavenging (i.e., the transition from oxide absorption to reducing burial), as the precipitation rates of Mo under reducing conditions are much higher than those of oxide absorption?

Line 222-225. Are all these Mo concentration data derived from highly oxic Mo sink (i.e., absorbed by Fe-Mn oxides)?

Line 253. Is high temperature hydrothermal fluid a Mo sink?

Fig. 1. Mo contents of GC112 core are quite high (up to 120 ppm). This value is generally from reducing or anoxic sediments as the authors also compiled here.

Version 1:

Reviewer comments:

Reviewer #1

(Remarks to the Author)

Congratulations on this important contribution. I am fully satisfied with the revisions and see no obstacle to publication.

Reviewer #2

(Remarks to the Author)

Review of the revised version of the manuscript NCOMMS-25-10252A "Revised Oceanic Mo Isotope Budget from Deep-Sea Pelagic Sediments" by Wang et al. submitted to Nature Communications.

Similar to my statement in the first review of this manuscript and based on the presented data, I think the proposed scenario of Mo transport from bottom water into the sediment via diffusion and subsequent adsorption onto Fe and Mn (hydro)oxides is a possible scenario. However concerning the here presented data two major concerns stay:

1) The influence on Fe oxides on Mo and higher Mo association within the Fe (hydr)oxide leach fractions

Here three observations from the authors data argue against this significant Fe influence of Mo:

a) If authigenic Fe input has a significant effect on Mo removal, why in Mo/Ti values between two cores behave similar to Mn/Ti ratios and not Fe/Ti (Fig. 2)? In core XT19 higher Fe/Ti values compared to core GC112 would have an impact in authigenic Mo enrichment (Mo/Ti). This is not the case. In the different leached fraction of MnO and Fe (hydro) oxide phases, Mo concentration in both leach fractions covary with Mn concentration and while a negative covariation is observed with Fe! Does this show that Mo is likely bound to Mn phases?

b) As stated in my last review, Co concentration covary with Mn and Mo concentrations within the sediments presented here suggesting Mn as primary Mo host as for Co

c) Plotting the different Mo, Mn and Fe concentration data from the different leaching steps (Table S2), there is a covariation between Mn and Mo concentrations within the Mn oxide phase and Fe (hydro) oxide phase sequential extraction. Although Mn concentration are quite low in second leach step, does this show that during both leaching steps the Mo is dominantly released from Mn-phases or that these different leaching steps were not quantitatively able to fully discriminate between both phases.?

2) Co/Mn covariation with Mo/Ti and $\delta^{98}\text{Mo}$ (Fig 3f)

The authors state in their response letter: "Because Mn and Co concentrations are significantly lower in seawater than in pore water, it is unlikely that bottom water influx influenced variations in the Co/Mn ratio." However the authors also mention in the manuscript (line 301) that "Co/Mn ratio (a proxy for geochemical accumulation rate) in deep-sea sediments (Fig. 3F), likely reflecting variations in Fe-Mn (hydro)oxide mineralogy and crystal structures formed under different depositional conditions".

There is a significant covariation between authigenic Mo enrichment (Mo/Ti ratios) and Co/Mn ratios. Especially in core GC112 which do show larger Mo/Ti variations with higher Mo/Ti ratios are leading to lower Co/Mn ratios ($R^2=0.85$). Is a systematic relationship between Co/Mn, $\delta^{98}\text{Mo}$ and Mo/Ti line with the suggested process of bottom water Mo infiltration into the sediment and subsequent cycling within the deeper sediment column or can this effect seen between Co/Mn, $\delta^{98}\text{Mo}$ and Mo/Ti be explained mineral transformation from birnessite to todorokite? (e.g. effect of mineral transformation of Ni isotopes (Fleischmann, EPSL, 2023))

Reviewer #3

(Remarks to the Author)

I think the authors have fully considered my previous comments and carefully revised the manuscript. I have no more comments this time, while only a small suggestion:

The figure S8 that shows comparing $\delta^{98}\text{Mo}$ values of bulk sediments and oxide phases is critical to demonstrate the burial process of Mo within the sediment profile. Thus, I recommend the authors move this figure to the main text of the manuscript, instead of the Supplementary Information.

Version 2:

Reviewer comments:

Reviewer #2

(Remarks to the Author)

The authors have considered my previous comments and addressed my concerns. Although I still have concerns about

the robustness of interpretation extracted from the sequential leaching data. But this will be potential subject to further future studies and therefore I have no more comments this time.

REVIEWER COMMENTS

Reviewer #1 (Remarks to the Author):

This is an important contribution to understanding the Mo and Mo isotope cycle that I think should definitely be published after minor to moderate revisions. The findings are provocative and have potentially huge implications for reinterpreting past paleoceanographic studies. The paper is generally well written and I see no issues with the data or interpretation although I really wish the authors would more fully take the plunge at the end and elaborate more on the implications for using Mo isotopes as a paleoceanographic redox tool. The authors seem to walk up to the edge of that ledge but I think the paper would be better if they took the plunge. Below are some line-by-line comments and suggestions.

We are very grateful to Reviewer #1 for providing constructive, insightful, and detailed reviews of our manuscript. The comments have significantly improved the quality of the work. Thank you for your time and patience.

In the revised manuscript, we have made the following changes:

The language or word choice in the first sentence is not great. Specifically, the phrase “dual role” doesn’t fit particularly well because you are talking about two totally unrelated things (i.e., “apples and oranges”). The first role is a role in biological processes whereas the second is in a completely unrelated topic: scientific inquiry.

Thank you for the valuable suggestion. We have revised the sentence on Lines 56–61 accordingly. In the revised manuscript, the sentence now reads:

“Molybdenum (Mo) serves as a powerful proxy for reconstructing paleoceanographic redox conditions through its abundance and isotopic variations”

Lines 80-82. It might be worth providing an additional sentence here to remark in more detail on WHAT the implications are.

Thank you for the valuable suggestion. The detail on the implications has been added (Lines 83-86). In the revised manuscript, we have changed the descriptions, as:

“The updated global Mo isotope mass balance model suggests that previous studies significantly overestimated the extent of euxinic seafloor in ancient oceans throughout geological history.”

Please indicate how you are reporting your $\delta^{98}\text{Mo}$ values early on in the main text of the paper. In other words, are you setting NIST3134 to 0 per mil (Goldberg) or adding a “+0.25” per Nagler.

Thank you for the valuable suggestion. There were errors, which had been made corrections (Lines 487-489). The Mo isotopic composition is reported as $\delta^{98}\text{Mo}$ values relative to NIST SRM 3134 +0.25‰, in accordance with the method established by Nögler et al. (2013). In the revised manuscript, we have changed the descriptions, as: “Here, all data, including those published previously, are reported relative to NIST SRM 3134+0.25‰⁵⁴.”

Lines 85-86. To me, it looks like the Mo/Al and especially Mn/Al of site XT19 do not really change with depth. Also, in the sentence beginning in line 85, it makes more sense to say “Mn/Al ratios and Mo/Al...” since this is the order that you show them in the data plots.

We thank the reviewer for their comments. As aluminum (Al) in these sediments is significantly influenced by authigenic components, we have replaced Al with titanium (Ti) for normalization. Ti serves as a more reliable indicator for quantifying detrital contributions in marine sediments (see Supplementary Materials S1, for further details). Accordingly, the description of the variations in Mn/Ti, Fe/Ti, and Mo/Ti ratios with depth has been revised (Lines 93–97). In the updated manuscript, the descriptions are as follows: “The Mn/Ti, Fe/Ti, and Mo/Ti ratios exhibit a progressive increase with depth from 0 to 4.0 mbsf, remaining stable in the deeper layers of core GC112, while showing a slight upward trend in core XT19 (Fig. 2)”.

Line 88 – Either get rid of the semicolon or get rid of the “whereas”

The original content has been deleted as suggested (Lines 96–99).

Lines 93-95. It’s quite striking how similar the highest and lowest values are in these two very distant localities!

We appreciate the reviewer’s insightful comments. The Mo isotopic compositions in these two cores displays intriguing patterns. Notably, Mo isotope data from a South Pacific core, published in Ahmad. (2021), also exhibited similar variation patterns and ranges. This similarity likely results from comparable early diagenetic processes in

these sediments, leading to analogous isotopic fractionation.

Relevant content has been added in the revised manuscript (Line 110-114), as:

“Similarly, Mo isotopes in the deep-sea sediments of the South Pacific region demonstrate a progressively increasing trend with depth from $-0.20 \pm 0.02\%$ to $0.44 \pm 0.02\%$ ¹⁹”

Ahmad Q, et al. The Molybdenum isotope subduction recycling conundrum: A case study from the Tongan subduction zone, Western Alps and Alpine Corsica. *Chem Geol* 576, 120231 (2021).

Line 96 – “patter” should be pattern

It has been modified (Line 108-110)

Line 97 – “ang” should be and

It has been modified (Line 110-112)

Line 98 – What do you mean by “bottom layer”?

Thank you for the valuable suggestion. It has been modified (Line 110-112)

Line 111–When you initially mention hydrothermal activity, I thought you meant general input of metals to oceanwater from hydrothermal vents but it sounds like what you are actually talking about is in situ deposition of metals within the sediment by hydrothermal seeps from directly below? Maybe clarify this.

Thank you for the valuable suggestion. It has been modified (Line 135-142). In the revised manuscript, we have changed the descriptions, as:

“Previous research indicates that hydrothermal activity significantly influences Fe and Mn concentrations in South and East Pacific sediments^{23, 24}. Hydrothermal fluids, introduced into seawater via vent sites or through diffuse flow from flanking regions, transport considerable quantities of Fe and Mn. Subsequent deposition of these metals alters the sediments' geochemical composition. In contrast, the Western Pacific sediments analyzed here contain markedly lower Fe/Ti (<14.0) and Mn/Ti (<6.0) (Fig. 2A–2B), suggesting minimal hydrothermal influence”

Section running from about Line 140 to 153 – If the Fe-Mn oxides higher in the sediment section are being reductively dissolved, does this say something about the redox gradient in the sediment package that might affect Mo solubility given that its behavior is quite different than Fe and Mn? In other words, would this inhibit Mo

movement deeper into the sediment pile? Or do you need outright sulfidic conditions for this to be an issue for Mo?

Thank you for the valuable suggestion. The redox state of the studied core was inferred using geochemical indicators from pore water in cores retrieved from adjacent regions. In the Western Pacific study area, bottom waters exhibit relatively high dissolved oxygen concentrations (Hondt et al., 2015; Chen et al., 2022). Pore water profiles throughout the core reveal nitrate and sulfate concentrations comparable to ambient seawater levels, while dissolved Fe and Mn concentrations remain very low (0.05–0.3 $\mu\text{mol/L}$) (Deng, 2021). These low concentrations contrast sharply with the significantly higher Fe (25–200 $\mu\text{mol/L}$) and Mn (16–180 $\mu\text{mol/L}$) levels characteristic of pore waters under nearshore reducing conditions (Goldberg et al., 2012). Collectively, this evidence suggests predominantly oxic conditions within the sediment core. Such conditions are consistent with previous studies demonstrating that modern bottom-water oxygen concentrations in deep-sea environments (typically >4,000 m depth) exceed 140 $\mu\text{mol/kg}$ (Garcia et al., 2019), and that oxygen may plausibly penetrate to the volcanic basement (Hans et al., 2012; Hondt et al., 2004, 2015). Relevant content has been added in the revised manuscript (Line 153-213).

In the revised manuscript, we have changed the descriptions, as:

“This observed pattern of increasing Mn/Ti and Fe/Ti ratios with depth aligns with previous findings in numerous deep-sea sediment cores^{6, 13, 15, 19, 20, 21, 22, 26, 27}. Two primary mechanisms have been proposed for this Mn and Fe enrichment: (1) reductive dissolution and subsequent re-oxidation, where Mn and Fe oxide reduction in pore water generates dissolved Mn and Fe ions that then re-oxidize and accumulate as oxides in deeper sediment layers²⁶; and (2) oxidative precipitation, involving direct Fe and Mn precipitation from the water column under oxidizing bottom-water conditions during sediment deposition. The latter mechanism is governed by the availability of Fe and Mn in bottom waters and the sediment accumulation rate.

Bottom waters in the Western Pacific study area exhibit high dissolved oxygen concentrations^{16, 28}. Pore water profiles from cores in adjacent regions show SO_4^{2-} and NO_3^- concentrations equal to or exceeding those of seawater. Dissolved Fe and Mn

concentrations are low (0.05–0.3 $\mu\text{mol/L}$)²⁹, contrasting sharply with the significantly higher Fe (25–200 $\mu\text{mol/L}$) and Mn (16–180 $\mu\text{mol/L}$) levels typical of pore waters in nearshore reducing environments³⁰. These values indicate that predominantly oxic conditions prevail within the sediment column. These conditions align with previous research indicating that modern bottom-water oxygen concentrations in deep-sea environments (typically >4,000 m depth) exceed 140 $\mu\text{mol/kg}$ ³¹, and that oxygen can penetrate to the volcanic basement^{28, 32, 33}. Consequently, enhanced precipitation under oxidizing bottom-water conditions represents the most plausible explanation for the observed Fe and Mn enrichment with depth. Notably, while the dissolved Fe and Mn concentrations in pore water from adjacent regions (0.05–0.3 $\mu\text{mol/L}$) are low compared to those in reducing environments, they exceed typical seawater levels. Research suggests that, under oxic conditions, such elevated dissolved Fe primarily originates from colloidal Fe released during the non-reductive diagenesis of tephra sediments^{34, 35, 36}. By analogy, the observed Mn enrichment may also largely result from the release of colloidal Mn via similar non-reductive diagenetic processes involving tephra material.

The Mo/Ti ratios in both sediment cores exhibit variation patterns similar to those of Mn/Ti and Fe/Ti (Fig. 2), suggesting that sedimentary Mo might co-precipitate with Fe and Mn oxides. However, if Mo were to deposit and enrich concurrently with Fe and Mn oxides, the Mn/Mo and Fe/Mo ratios in the sediment, along with its Mo isotopic composition, should align with values from the top surface of both sediment cores and remain constant with depth. In contrast, our data show that Mn/Mo and Fe/Mo ratios gradually decrease with depth, while Mo isotopes become progressively heavier. Consequently, we hypothesize that post-sedimentation, sedimentary Mo is likely influenced by early diagenetic processes, such as the infiltration and addition of Mo from bottom waters. This hypothesis aligns with previous findings that early diagenetic processes are primary factors influencing the fractionation of rare earth elements and Zn isotope composition in Pacific Ocean deep-sea sediments^{23, 29}. This is further corroborated by the observation that dissolved Mo concentrations in the pore waters of Western Pacific sediments are generally lower than bottom water concentrations and

decrease with depth (from approximately 110 nM—the seawater Mo concentration—at the sediment-water interface to about 20 nM at 15 meters below seafloor (mbsf)³⁷). This indicates continuous Mo transport from bottom water into the sediment via diffusion and subsequent adsorption onto Fe and Mn oxides. Due to higher Mn and Fe content and/or prolonged adsorption in deeper layers, these sediments exhibit greater Mo enrichment than shallower layers. This leads to enhanced Mo accumulation at depth (Fig. 2C and Fig. S2A), and Mo/Ti ratios show a positive correlation with Mn/Ti and Fe/Ti ratios (Fig. 3A and Fig. 1B). Conversely, Fe and Mn concentrations in pore water are slightly higher than in bottom water, which precludes their sustained transport from bottom water into sediments. Ultimately, these processes lead to a gradual decrease in the Mn/Mo and Fe/Mo ratios with depth^{19, 20, 21}.”

Line 208-210 – The crusts form slowly but how do the Mo concentrations in these compare to the sediments?

Thank you for the valuable suggestion. The comparison method specifically involves evaluating output fluxes. Based on the deposition rates of manganese nodules (Mn), previous studies have demonstrated that the output flux of Mo ranges from 0.09 to $0.35 \times 10^8 \text{ mol yr}^{-1}$, while the flux estimated from sediment deposition rates is approximately $0.87 \times 10^8 \text{ mol yr}^{-1}$. Thus, the former accounts for 10% to 41% of the latter (Little et al., 2025). Additionally, compared to deep-sea sediments, the areal extent and volume of iron-manganese nodules on the global ocean floor are relatively restricted. Finally, deep-sea sediments have traditionally been employed as end-members for marine oxidation processes when estimating the oceanic Mo flux (Scott, 2008). As such, they can be reasonably considered end-members for the Mo isotope composition within marine oxidation environments.

This sentence has been changed in the revised manuscript (Line 320-324).

In the revised manuscript, we have changed the descriptions, as:

“While Fe-Mn crusts and nodules have traditionally represented oxic sedimentary Mo isotopes^{1, 8, 12}, their slow accumulation rates and relatively limited areal and volumetric distribution on the global ocean floor minimize their contribution to oceanic Mo removal¹²”

Paragraph starting at line 285 – It would be great if you could elaborate on the implications for previous interpretations of Mo isotope data in ancient sediments! Just one example: what bearing does your finding have on debates over the redox state of the Proterozoic oceans?

Thank you for the valuable suggestion. In this study, the Mo mass balance model (Chen et al., 2015; Wei et al., 2021) (see Supplementary Materials S5 for a detailed model description and parameters) was used to evaluate the impact of changes in the Mo isotopic composition of the oxic sink endmember on the reconstructed relative sizes of global euxinic seafloor, as inferred from marine Mo isotopes. The evaluations focused on five distinct ancient ocean Mo isotope compositions: three from key geological intervals—the late Paleoproterozoic (1.2‰), the Early Jurassic (1.6‰), and the Paleocene-Eocene Thermal Maximum (PETM) (2.0‰)—and two hypothetical values (1.4‰ and 1.8‰) (Kendall et al., 2011; Pearce et al., 2008; Dickson et al., 2012). The results demonstrate that when the oxic sink endmember changes—specifically, when the fractionation factor shifts from 3‰ to 2.4‰—there are substantial alterations in the calculated relative sizes of global euxinic seafloor areas across these intervals. Assuming that the relative sizes and areas of oxic seafloor are fixed, the inferred relative sizes increase significantly compared to previous estimates in Fig. 2. Relevant content has been added in the revised manuscript (Manuscript-marked Line 415-431) and (Supplementary Materials-marked Line 150-170).

In the revised manuscript, we have changed the descriptions, as:

“To evaluate the impact of these revisions, the Mo mass balance model (Supplementary Materials S5 for a detailed model description and parameters) was employed^{6, 9}. This model assessed how changes in the Mo isotopic composition of the oxic sink end-member affect reconstructed estimates of global euxinic seafloor area, as inferred from ancient ocean Mo isotope composition. The evaluations focused on five distinct ancient ocean Mo isotope compositions: three from key geological intervals—the late Paleoproterozoic (1.2‰)⁸, the Early Jurassic (1.6‰)⁵¹, and the Paleocene-Eocene Thermal Maximum (PETM) (2.0‰)⁵²—and two hypothetical values (1.4‰ and 1.8‰). The results demonstrate substantial alterations in the calculated relative extents of

global euxinic seafloor across these intervals when the oxic sink end-member is modified—specifically, when the fractionation factor ($\Delta^{98}\text{Moseawater-oxic sink}$) shifts from 3.0‰ to 2.40‰ (Fig. 5). Assuming the relative area of oxic seafloor [$F_{\text{OX}}/(F_{\text{OX}} + F_{\text{RED}})$] is held constant in Fig. 5, the reconstructed relative extent of global euxinic seafloor is less than previously indicated^{8, 51, 52}. In other words, prior estimates appear to have significantly overestimated the relative extent of euxinic seafloor.”

“Assuming a steady-state marine molybdenum (Mo) cycle, the $\delta^{98}\text{Mo}$ signature of global seawater predominantly reflects the partitioning of Mo among sediments deposited under three prevailing redox conditions: euxinic, anoxic (or suboxic), and oxic environments (Chen et al., 2015; Wei et al., 2021). When riverine input is regarded as the principal source of oceanic Mo, its flux is typically standardized to modern values for consistency. Under these conditions, the isotopic composition of seawater Mo is determined by the relative fluxes to each redox sink, as described by the relationship:

$$F_{\text{EUX}} + F_{\text{RED}} + F_{\text{OX}} = F_{\text{Rivers}}$$

The isotopic mass balance is expressed as:

$$\delta^{98}\text{Mo}_{\text{river}} = F_{\text{OX}} \times (\delta^{98}\text{Mo}_{\text{SW}} + \Delta_{\text{OX}}) + F_{\text{RED}} \times (\delta^{98}\text{Mo}_{\text{SW}} + \Delta_{\text{RED}}) + F_{\text{EUX}} \times (\delta^{98}\text{Mo}_{\text{SW}} + \Delta_{\text{EUX}})$$

where F_{OX} , F_{RED} , and F_{EUX} represent Mo output flux to each sediment type. $\delta^{98}\text{Mo}_{\text{river}}$ and $\delta^{98}\text{Mo}_{\text{SW}}$ denote the isotopic compositions of the riverine input and seawater, respectively, while Δ refers to the isotopic fractionation between seawater and each sedimentary flux (OX = oxic sink; RED = reducing sink; EUX = euxinic sink). The necessary parameters, including the Mo fluxes for sediment output and riverine input, as well as fractionation magnitudes, are provided for the modern ocean (Table S4). It is worth emphasizing that $\delta^{98}\text{Mo}_{\text{river}}$ is provisionally set at 0.7‰ (Kendall et al., 2017).

To obtain $\delta^{98}\text{Mo}_{\text{SW}}$ as a function of the removal fluxes (F_{OX} , F_{RED} , and F_{EUX}), we solve the equations above and subsequently plot the contours of $\delta^{98}\text{Mo}_{\text{SW}}$ in Fig. 5.”

Figure 1. The data plots really need larger text type. Also, why not have Mo on the y-axis in both of the data plots?

Thank you for the valuable suggestion. We improved the clarity and precision of Figure 1 by redrawing it, increasing the font size, and recalibrating the coordinate axes.

Line 548-550 – There are no light purple bars in this figure. Also, the last two sentences in the figure caption seem to be saying precisely the same thing; maybe you forgot to delete one of them before submitting this?

Thank you for the valuable suggestion, we have remodified the caption.

Figure S3. So do the A, B, C, and D within each plot (marked on the horizontal plane) stand for phosphate, Mn oxide, Fe h-oxide, and residual? This should be spelled out in the caption.

Thank you for the valuable suggestion, it's have been spelled out in the caption in Figure S3.

Figure S4. These would be interesting diagrams, but the text components are small to the point of being unreadable.

Thank you for the valuable suggestion. The original figure was separated into two distinct figures: Fig. S4A and Fig. S4B.

Reviewer #2 (Remarks to the Author):

Review of the manuscript NCOMMS-25-10252 “Revised Oceanic Mo Isotope Budget from Deep-Sea Pelagic Sediments” by Wang et al. submitted to Nature Communications.

This manuscript present stable Mo isotope composition from two Pacific Ocean deep-sea pelagic sediment cores. Cores do show trend with increasing Mo isotope ratios with depth and covariation with Mo/Al with Mo/Al and Mn/Mo ratios. The authors argue that the depth-dependent increase in Mo isotope signatures are likely results of an initial preferential adsorption of lighter Mo isotopes during bottom water infiltration coupled

by adsorption of heavier Mo isotopes at greater depths. Based on the presented data I think this is a possible scenario and explanation for the increasing Mo isotope signatures with increasing core depth. I also have no doubt that the data presented here are of good quality. However, I have some small but also some more substantial comments concerning the influence of Mn diagenesis on the Mo isotopes composition, the use of Co as tracer geochemical accumulation rate and influence of Fe-oxides on Mo in these sediments. These require revision.

We are very grateful to Reviewer #2 for their insightful and helpful comments. Their suggestions have significantly improved our manuscript. Thank you very much for your valuable feedback!

Line 85: "...in both sediment cores. Mo/Al ratios and Mn/Al ratios (or concentrations) increase with..." Please specify the "or concentration". Ratios or concentrations?

We thank the reviewer for their comments. As aluminum (Al) in these sediments is significantly influenced by authigenic components, we have replaced Al with titanium (Ti) for normalization. Ti serves as a more reliable indicator for quantifying detrital contributions in marine sediments (see Supplementary Materials S1, for further details). Accordingly, the description of the variations in Mn/Ti, Fe/Ti, and Mo/Ti ratios with depth has been revised (Lines 93–97). In the updated manuscript, the descriptions are as follows: "The Mn/Ti, Fe/Ti, and Mo/Ti ratios exhibit a progressive increase with depth from 0 to 4.0 mbsf, remaining stable in the deeper layers of core GC112, while showing a slight upward trend in core XT19 (Fig. 2)"

Line 87: "in XT19, they show a continuous increase with depth;" Considering Fig.2F it is not an continuous increase. There are quite some fluctuation the first 2 m down-core.

We thank the reviewer for their comments. As aluminum (Al) in these sediments is significantly influenced by authigenic components, we have replaced Al with titanium (Ti) for normalization. Ti serves as a more reliable indicator for quantifying detrital contributions in marine sediments (see Supplementary Materials S1, for further details). Accordingly, the description of the variations in Mn/Ti, Fe/Ti, and Mo/Ti ratios with depth has been revised (Lines 93–97). In the updated manuscript, the descriptions are

as follows: “The Mn/Ti, Fe/Ti, and Mo/Ti ratios exhibit a progressive increase with depth from 0 to 4.0 mbsf, remaining stable in the deeper layers of core GC112, while showing a slight upward trend in core XT19 (Fig. 2)”

Line 87: “Core GC112 demonstrates a monotonic increase in $\delta^{98}\text{Mo}$ values with depth...I agree with the authors that there is an increase in $\delta^{98}\text{Mo}$ values with depth, but it is not monotonic.

Thanks to the reviewer's comments. It has been modified (Line 105-108). In the revised manuscript, we have changed the descriptions, as: “Overall, $\delta^{98}\text{Mo}$ values in Core GC112 exhibit a gradual increase with depth, ranging from $-0.55 \pm 0.04\%$ to $0.19 \pm 0.03\%$, interspersed with minor decreases at several depths (e.g., 2.5 mbsf, 4.2 mbsf, and 6.0 mbsf)”

Line 97: “...0.03‰ at 3.0 mbsf, ang then decrease to 0.50‰ at the bottom layer”

Please correct “ang”

Thank you very much. It has been modified (Line 110-112)

Line 98: “the citation number 19”

Indeed, as the authors state the Goldschmidt abstract, they cite show an increase in $\delta^{98}\text{Mo}$ with burial depth. However, these authors state that the positive relation between Mo content and isotope composition indicates that the Mo carrier (i.e., Fe-Mn (oxyhydr)oxides) might preferentially release the isotopically heavy Mo during Mn reduction.

On the other hand, another published article where some authors of this submitted manuscript are listed co-authors “Extremely light molybdenum isotope signature of sediments in the Mariana Trench” published in Chemical Geology from Chen et al 2022. There extremely light $\delta^{98}\text{Mo}$ down to $-1.71 \pm 0.05 \%$. Here light $\delta^{98}\text{Mo}$ values of sediments in deeper parts of the cores coincide with enhanced Mo accumulation. Here the authors state that the source of the isotopically light dissolved Mo was explained by the preferential release of isotopically lighter Mo from Fe-Mn (oxyhydr)oxides during the enhanced dissolution at deeper depth.

So both, the cited abstract and Chen et al 2022 state that Fe-Mn (oxyhydr)oxides diagenesis/reduction has an influence on the sediment $\delta^{98}\text{Mo}$ pattern along a core

profile.

Thank you for the reviewers' constructive comments. The abstract content has not yet been subject to peer review, and the proposed mechanism represents a preliminary hypothesis requiring further substantiation, particularly through additional evidence such as pore water data. Several aspects of the described mechanism warrant further discussion. First, if Mo isotopic variations above 3 meters below seafloor (mbsf) are influenced by the upward diffusion of isotopically heavier Mo from depths below 3 mbsf, then Mo concentrations in the upper section should theoretically decrease towards the sediment surface, rather than remain constant. Second, existing research indicates that surface sediments are typically dominated by Mo adsorption onto Fe-Mn (oxyhydr)oxides, resulting in isotopic signatures near -0.75‰, similar to ferromanganese nodules. If the isotopic composition above 3 mbsf is indeed affected by upward diffusion from below, the surface sediment's isotopic signature would not be expected to solely reflect this characteristic adsorption process. Finally, the implication of continuous Mo diffusion from the sediment core into the overlying bottom water suggests that deep-sea sediments act as a significant Mo source over various timescales, rather than a sink. This assertion contradicts the currently prevailing understanding of marine Mo cycling.

Chen et al. (2022) proposed that the preferential release of isotopically lighter Mo from Fe-Mn (oxyhydr)oxides during enhanced dissolution at greater depths (exceeding the depth of four sediment cores in this study) explains the extremely light Mo isotopic compositions observed in deep-sea sediments. Ahmad et al. (2021) employed the mechanism of light Mo isotope release via the reduction of Fe-Mn (oxyhydr)oxides to explain the observed increase in Mo isotopic values (enrichment in heavier isotopes) with depth in deep-sea sediments. However, these mechanisms do not adequately explain the Mo isotope variations identified in our study.

The corresponding content has been added in the manuscript (Line 270-296). In the revised manuscript, we have changed the descriptions, as:

“Chen et al. (2022)¹⁶ attributed exceptionally light Mo isotopic compositions in deep-

sea sediments to the preferential release of lighter Mo isotopes from Fe-Mn (oxyhydr)oxides during enhanced dissolution at depths exceeding those they investigated. Indeed, in cores GC112 and XT19 above 4.0 mbsf, our data show a consistent decrease in Mn/Ti, Fe/Ti, and Mo/Ti ratios toward the sediment surface (Fig. 2A-2C). However, if Fe-Mn (oxyhydr)oxide dissolution drove this reduction, as Chen et al. (2022)¹⁶ proposed, sedimentary Mo isotopes should become progressively heavier (i.e., exhibit increasing $\delta^{98}\text{Mo}$ values) upwards from 4.0 mbsf, owing to the preferential release of lighter isotopes. Contrary to this expectation, our data reveal a gradual enrichment in lighter Mo isotopes (decreasing $\delta^{98}\text{Mo}$ values) upwards from this depth (Fig. 2D). Therefore, this proposed mechanism does not adequately explain the Mo isotopic variations observed in our study.

Similarly, Ahmad et al. (2021)¹⁹ attributed the observed increase in Mo isotopic values (signifying an enrichment in heavier isotopes) with depth in deep-sea sediments to the preferential release of light Mo isotopes during the reductive dissolution of Fe-Mn (oxyhydr)oxides. They posited that with increasing sediment depth, this process preferentially releases lighter Mo isotopes, leading to a progressive decrease in solid-phase Mo content and a concomitant enrichment of heavier isotopes in the residual sediment. Consequently, a strong negative correlation between Mo concentration and its isotopic composition would be anticipated. In contrast, our core samples exhibit contrary trends: Mo/Ti ratios (or Mo concentrations) increase with depth (Fig. 3C and Fig. S2A), accompanied by a positive correlation between Mo/Ti ratios and isotopic composition (Fig. 3C). Therefore, the mechanism proposed by Ahmad et al. (2021)¹⁹

also fails to adequately explain the Mo concentration and isotopic signatures observed in our study.”

Line 128: “Previous studies have documented increasing Mn concentrations with depth in pelagic sediments, attributed to progressive Mn oxidation in porewaters leading to enhanced Mn oxide accumulation in deeper sediment layers.”

For citation 28 from Kato et al 2011 “Deep-sea mud in the Pacific Ocean as a potential resource for rare - earth elements”, I was not able to find such a statement for increasing Mn concentrations with depth.

Thank you for the reviewers’ constructive comments. An interesting phenomenon observed is that Mn/Ti and Fe/Ti ratios in many deep-sea sediment cores increase with depth (Bi et al., 2021; Tanaka et al., 2020, 2023). The corresponding content have been added in the manuscript (Manuscript-marked: Line 153-180).

In the revised manuscript, we have changed the descriptions, as: “This observed pattern of increasing Mn/Ti and Fe/Ti ratios with depth aligns with previous findings in numerous deep-sea sediment cores^{6, 13, 15, 19, 20, 21, 22, 26, 27}. Two primary mechanisms have been proposed for this Mn and Fe enrichment: (1) reductive dissolution and subsequent re-oxidation, where Mn and Fe oxide reduction in pore water generates dissolved Mn and Fe ions that then re-oxidize and accumulate as oxides in deeper sediment layers²⁶; and (2) oxidative precipitation, involving direct Fe and Mn precipitation from the water column under oxidizing bottom-water conditions during sediment deposition. The latter mechanism is governed by the availability of Fe and Mn in bottom waters and the sediment accumulation rate.

Bottom waters in the Western Pacific study area exhibit high dissolved oxygen concentrations^{16, 28}. Pore water profiles from cores in adjacent regions show SO_4^{2-} and NO_3^- concentrations equal to or exceeding those of seawater. Dissolved Fe and Mn concentrations are low (0.05–0.3 $\mu\text{mol/L}$)²⁹, contrasting sharply with the significantly higher Fe (25–200 $\mu\text{mol/L}$) and Mn (16–180 $\mu\text{mol/L}$) levels typical of pore waters in nearshore reducing environments³⁰. These values indicate that predominantly oxic conditions prevail within the sediment column. These conditions align with previous

research indicating that modern bottom-water oxygen concentrations in deep-sea environments (typically >4,000 m depth) exceed 140 $\mu\text{mol}/\text{kg}^{31}$, and that oxygen can penetrate to the volcanic basement^{28, 32, 33}. Consequently, enhanced precipitation under oxidizing bottom-water conditions represents the most plausible explanation for the observed Fe and Mn enrichment with depth.”

References

- Bi, D., Shi, X., Huang, M., Yu, M., Zhou, T., Zhang, Y., Zhu, A., Shi, M., and Fang, X., 2021, Geochemical and mineralogical characteristics of deep-sea sediments from the western North Pacific Ocean: Constraints on the enrichment processes of rare earth elements: *Ore Geology Reviews*, v. 138, p. 104318, <https://doi.org/10.1016/j.oregeorev.2021.104318>.
- Tanaka, E., Mimura, K., Nakamura, K., Ohta, J., Yasukawa, K., and Kato, Y., 2023, Rare-Earth Elements in Deep-Sea Sediments in the South Pacific Gyre: Source Materials and Resource Potentials: *Geochemistry Geophysics Geosystems*, v. 24, no. 3, p. 2022GC010681, <https://doi.org/10.1029/2022gc010681>.
- Tanaka, E., Nakamura, K., Yasukawa, K., Mimura, K., Fujinaga, K., Iijima, K., Nozaki, T., and Kato, Y., 2020, Chemostratigraphy of deep-sea sediments in the western North Pacific Ocean: Implications for genesis of mud highly enriched in rare-earth elements and yttrium: *Ore Geology Reviews*, v. 119, p. 103392, <https://doi.org/10.1016/j.oregeorev.2020.103392>.
- Yasukawa, K., Liu, H., Fujinaga, K., Machida, S., Haraguchi, S., Ishii, T., Nakamura, K., and Kato, Y., 2014, Geochemistry and mineralogy of REY-rich mud in the eastern Indian Ocean: *Journal of Asian Earth Sciences*, v. 93, p. 25-36, <https://doi.org/10.1016/j.jseaes.2014.07.005>.
- Chen, S. et al., 2022. Extremely light molybdenum isotope signature of sediments in the Mariana Trench. *Chemical Geology*, 605: 120959.
- Brucker, R., McManus, J., Severmann, S., Berelson, W., 2009. Molybdenum behavior during early diagenesis: Insights from Mo isotopes. *Geochemistry Geophysics Geosystems*, 10(6): Q06010.
- Bertine, K.K., Turekian, K.K., 1973. Molybdenum in Marine Deposits. *Geochimica et Cosmochimica Acta*, 37(6): 1415-1434.

Line 154: “A key finding in this study is the progressively heavier Mo isotopic signature observed in the deep-sea sediments with increasing depth, contrasting with the lighter signatures found in Fe-Mn nodules and crusts (Fig. 2C).”

I think it is fair to mention that this finding has also been discovered by Ahmad et al 2021 for a Mn- rich pacific ocean drill core “Decreasing Mo/Mn and higher $\delta^{98/95}\text{Mo}$ of sediments with core depth and depositional age potentially record an additional early diagenetic process” However this has been interpreted to “resulting in Mo loss and fractionation likely related to the aging of Mn-oxyhydroxides during sediment deposition and compaction, which causes a higher structural order and lower reactivity of the Mn-phases.”

This is different to the interpretation presented here.

Thank you for the reviewers’ constructive comments. We have now cited the referenced article and examined its proposed mechanism in our discussion (Line 110-114 and Lines 270-295). In the revised manuscript, the text now reads:

“Similarly, Mo isotopes in the deep-sea sediments of the South Pacific region demonstrate a progressively increasing trend with depth from $-0.20 \pm 0.02\text{‰}$ to $0.44 \pm 0.02\text{‰}$ ¹⁹”

“Chen et al. (2022)¹⁶ attributed exceptionally light Mo isotopic compositions in deep-sea sediments to the preferential release of lighter Mo isotopes from Fe-Mn (oxyhydr)oxides during enhanced dissolution at depths exceeding those they investigated. Indeed, in cores GC112 and XT19 above 4.0 mbsf, our data show a consistent decrease in Mn/Ti, Fe/Ti, and Mo/Ti ratios toward the sediment surface (Fig. 2A-2C). However, if Fe-Mn (oxyhydr)oxide dissolution drove this reduction, as Chen et al. (2022)¹⁶ proposed, sedimentary Mo isotopes should become progressively heavier (i.e., exhibit increasing $\delta^{98}\text{Mo}$ values) upwards from 4.0 mbsf, owing to the preferential release of lighter isotopes. Contrary to this expectation, our data reveal a gradual enrichment in lighter Mo isotopes (decreasing $\delta^{98}\text{Mo}$ values) upwards from this depth (Fig. 2D). Therefore, this proposed mechanism does not adequately explain the Mo isotopic variations observed in our study.

Similarly, Ahmad et al. (2021)¹⁹ attributed the observed increase in Mo isotopic values (signifying an enrichment in heavier isotopes) with depth in deep-sea sediments to the preferential release of light Mo isotopes during the reductive dissolution of Fe-Mn (oxyhydr)oxides. They posited that with increasing sediment depth, this process preferentially releases lighter Mo isotopes, leading to a progressive decrease in solid-phase Mo content and a concomitant enrichment of heavier isotopes in the residual sediment. Consequently, a strong negative correlation between Mo concentration and its isotopic composition would be anticipated. In contrast, our core samples exhibit contrary trends: Mo/Ti ratios (or Mo concentrations) increase with depth (Fig. 3C and Fig. S2A), accompanied by a positive correlation between Mo/Ti ratios and isotopic composition (Fig. 3C). Therefore, the mechanism proposed by Ahmad et al. (2021)¹⁹ also fails to adequately explain the Mo concentration and isotopic signatures observed in our study.”

Line 189: “Our research reveals a negative correlation between $\delta^{98}\text{Mo}$ values and the Co/Mn ratio (a proxy for geochemical accumulation rate) in deep-sea sediments (Fig. 5),...” A reference is needed for Co/Mn as proxy for geochemical accumulation rate. For me this is not conclusive: Co concentration correlate with Mn and Mo concentrations ($R=0.89$ between Mo and Co). I somehow expect these since Co and Mo both are enriched in Mn-oxides. However, as Co shows the same concentration pattern as Mn and Mo and both are affected by diagenesis and/or migration through pore-water, how Co/Mn ratios can be a proxy for geochemical accumulation rate.

We appreciate the reviewers' insightful feedback. Geochemical analysis of pore water from sediments in adjacent marine areas reveals predominantly oxic conditions throughout the studied core. This finding indicates that the observed manganese (Mn) and cobalt (Co) enrichment is primarily due to in-situ deposition under these oxygen-

rich conditions. Because Mn and Co concentrations are significantly lower in seawater than in pore water, it is unlikely that bottom water influx influenced variations in the Mn/Co ratio. Previous research has established that the Co flux to sediments is nearly constant over time and area (Halbach et al., 1983), meaning Co concentration in Fe-Mn deposits is inversely related to the accumulation rate (e.g., Manheim and Lane-Bostwick, 1988). Consequently, the Co/Mn ratio may serve as a viable proxy for estimating deposition rates during the investigated period (Manheim et al., 1998; Fleischmann et al., 2023; Zhang et al., 2024).

References

Fleischmann S, et al. The nickel output to abyssal pelagic manganese oxides: A balanced elemental and isotope budget for the oceans. *Earth Planet Sc Lett* 619, 118301 (2023).

Zhang GL, et al. Balancing the oceanic Zn isotope budget: The key role of deep-sea pelagic sediments. *Geology* 52, 789-793 (2024).

Manheim FT, Lane-Bostwick CM. Cobalt in ferromanganese crusts as a monitor of hydrothermal discharge on the Pacific sea floor. *Nature* 335, 59-62 (1988).

Halbach P, Segl M, Puteanus D, Mangini A. Co-fluxes and growth rates in ferromanganese deposits from central Pacific seamount areas. *Nature* 304, 716-719 (1983).

Line 209: “However, existing Mo isotope data for deep sea sediments remains limited, with previous Pacific Ocean studies focusing solely on surface layers (<30 cm)”

Not quite right, see Ahmad et al 2021

We appreciate the reviewers' constructive comments. We acknowledge that the omission of this article led to an inaccurate statement, which we have now corrected (Lines 326-328).

Line 237-246 “Our samples, representing only ~7.5m of deep-sea sediments, likely.... Consequently, the mean $\delta^{98}\text{Mo}$ of the entire deep-sea sediment column is likely higher than $-0.09 \pm 0.23\%$ and significantly exceeds the $\delta^{98}\text{Mo}$ observed in Fe and Mn oxide crusts and nodules ($-0.70 \pm 0.10\%$). This observation is supported by Ahmad et al 2021 also for sediment below 7.5 m and should be cited here.

Thank you for the reviewers' constructive comments. We have now cited the referenced

article and examined its proposed mechanism in our discussion (Line 110-114 and Lines 282-295). In the revised manuscript, the text now reads:

“Similarly, Mo isotopes in the deep-sea sediments of the South Pacific region demonstrate a progressively increasing trend with depth from $-0.20 \pm 0.02\%$ to $0.44 \pm 0.02\%$ ¹⁹”

“Similarly, Ahmad et al. (2021)¹⁹ attributed the observed increase in Mo isotopic values (signifying an enrichment in heavier isotopes) with depth in deep-sea sediments to the preferential release of light Mo isotopes during the reductive dissolution of Fe-Mn (oxyhydr)oxides. They posited that with increasing sediment depth, this process preferentially releases lighter Mo isotopes, leading to a progressive decrease in solid-phase Mo content and a concomitant enrichment of heavier isotopes in the residual sediment. Consequently, a strong negative correlation between Mo concentration and its isotopic composition would be anticipated. In contrast, our core samples exhibit contrary trends: Mo/Ti ratios (or Mo concentrations) increase with depth (Fig. 3C and Fig. S2A), accompanied by a positive correlation between Mo/Ti ratios and isotopic composition (Fig. 3C). Therefore, the mechanism proposed by Ahmad et al. (2021)¹⁹ also fails to adequately explain the Mo concentration and isotopic signatures observed in our study.”

Does this 2% represent the fraction of pelagic clays relative to the GLOSS? Is this true why they represent the uppermost 2%? Deep-sea pelagic sediments make up only a small portion of the global subducting sediment GLOSS (See Plank, The Chemical Composition of Subducting Sediments, 2014), with Sediment thickness for dominant Pelagic clay lithologies reaching ~70m (see Tonga).

Thank you for your insightful feedback. The previous paragraph has been removed and

replaced with a new section that explains why deep-sea sediments can function as end-members for molybdenum isotope values representing the oxic sink in the modern ocean (Lines 350–366). In the revised manuscript, the text now reads: “Deep-sea sediments, as integral components of the modern ocean system, play a vital role in regulating marine geochemical cycles through early diagenetic processes. Previous research indicates that seawater infiltration rates—a key driver of early diagenesis in deep-sea settings—are relatively high, ranging from 1.2 to 10 cm/year^{49, 50}. Based on the core depths analyzed in this study, the estimated timescale for seawater infiltration spans several decades to several hundred years. This duration is substantially shorter than the oceanic Mo residence time of 0.44 Ma¹². Furthermore, the global oceanic Mo isotopic composition, reconstructed from Fe-Mn nodules since the Cenozoic era, has remained stable¹. Consistent with this stability, Mo isotope compositions in the studied cores have undergone only minor changes since their deposition approximately 10 million years ago (Fig. S5). Collectively, these observations suggest that the exchange of Mo isotopes between deep-sea sediments and the ocean has largely been in a stable equilibrium state since the time of sediment deposition. Therefore, the average $\delta^{98}\text{Mo}$ value ($-0.09 \pm 0.23\%$) derived from these deep-sea sediments provides a reliable proxy for the oxic sink value in the modern ocean.”

Line 268: “Analysis reveals that variations in F_{RED} and $\delta^{98}\text{Mo}_{\text{RED}}$ substantially influence the total error in the equilibrium calculation, calculated as the absolute sum of the Mo mass balance and isotopic balance errors. In contrast, F_{EUX} and $\delta^{98}\text{Mo}_{\text{EUX}}$ exert minimal influence.”

It appears that the reviewer did not fully address this issue. We infer that the reviewer may have intended to suggest that it could be necessary to elaborate on the basis for how these parameters (e.g., F_{RED} and $\delta^{98}\text{Mo}_{\text{RED}}$) influence the total error.

We changed this sentence to the Supplementary Materials-marked (Lines 144-149). “Analysis of the equilibrium calculations indicates a moderate negative correlation between the total error and both F_{RED} ($R^2 = 0.15$, $p < 0.001$) and $\delta^{98}\text{Mo}_{\text{RED}}$ ($R^2 = 0.69$, $p < 0.001$). The total error is defined as the absolute sum of the errors in the Mo mass balance and isotopic balance. In contrast, neither F_{EUX} nor $\delta^{98}\text{Mo}_{\text{EUX}}$ shows a significant

linear relationship with the total error. These findings suggest that variations in F_{RED} and $\delta^{98}M_{ORED}$ are the primary contributors to the total error in the equilibrium calculation, while F_{EUX} and $\delta^{98}M_{OEUX}$ have a negligible impact.”

Fig S6, where the Parameter Distributions are shown the y-axis displays Mo isotope ratios in ‰ and the range of fluxes in 10^8 mol yr^{-1} – this should be better indicated.

We appreciate the reviewers' constructive comments. It has been added in the Fig. S6.

Another question regarding the position of the optimal solution (star) in Fig S6. For me the total Error in minimal for a large range of $\delta^{98}M_{OEUX}$ and F_{EUX} . So how the optimal solution (star) in Fig S6 has been constrained?

We appreciate the reviewers' constructive comments.

The optimal solution, marked by a star in Figure S6, is determined by the criterion of minimizing total error. Specifically, the total error is the sum of the Mass Balance Error and the Isotope Balance Error ($0.003312+0.000754=0.004066$). This value (0.004066), representing the minimum total error identified over 100,000 iterations, establishes this parameter set as optimal.

The calculations for the Mo mass balance and isotopic balance errors are performed as follows:

Mo mass balance Error Calculation:

$$\text{Mass Balance}=1.74-1.4528-0.2839=0.003312$$

$$\text{Mass Balance Error}=|0.003312|=0.003312$$

Isotope Balance Error Calculation:

$$\text{Input Terms}=(3.1\times 0.80)+(0.26\times 0.80)=2.688$$

$$\text{Output Terms}=[1.52\times(0.09)\times 1.9048\times 1.6061]+1.4528+0.1+[0.2839\times(0.5)]=-0.1368+0.5409+2.3334-0.05=2.6875$$

$$\text{Isotope Balance}=2.688-2.6875=0.0005$$

$$\text{Isotope Balance Error}\approx 0.000754.$$

The corresponding content have been added in the Supplementary Materials (Lines 143-148). In the revised manuscript, the text now reads:

“The optimal parameter set, identified by minimizing the total error (the sum of Mo mass balance and isotopic balance errors), achieves a minimum total error of 0.004066

over 100,000 iterations. These optimal values are $F_{\text{RED}} = 1.45 \times 10^8 \text{ mol yr}^{-1}$, $F_{\text{EUX}} = 0.28 \times 10^8 \text{ mol yr}^{-1}$, $\delta^{98}\text{M}_{\text{ORED}} = 1.61\%$, and $\delta^{98}\text{M}_{\text{OEUX}} = 1.90\%$, respectively (Fig. S6 and Table S3).”

Line 286: “While Mn oxides have traditionally been regarded as the primary form of Mo in marine oxidizing sediments, our research demonstrates that Fe oxides represent another critical form, accounting for over half of Mo occurrences (Fig. S3)”

This comes a little bit of surprise for me. Not that I do not agree that Fe-oxides can have an influence especially in setting with abundant Fe-oxide formation. However, calculating the authigenic Fe and Mn fraction within the sediment using the major element UCC concentrations from Rudnick & Gao as detrital input, as it was used for Mo by the authors, e.g. 80-90% of the Mn is authigenic compared to only 20-30% of Fe. Accounting the higher. A high correlation coefficient of 0.92 and 0.65 and between authigenic Mo and Mn concentration for core GC112 and TX12 respectively, indic

We appreciate the reviewers' constructive comments. Indeed, the proportions of authigenic Fe) and Mn in deep-sea sediments differ markedly, with Fe comprising 20-30% and Mn accounting for 80-90%. Although the proportions of authigenic components exhibit considerable variation, the absolute concentrations of authigenic Fe (1.0-2.0%) and Mn (0.5-1.0%) remain comparable, reflecting Fe's substantially higher concentration relative to Mn in deep-sea sediments. Mo adsorption experiments using ferrihydrite—a key (oxyhydr)oxide that typically constitutes 50-70% of the total Fe (oxyhydr)oxide pool in most marine environments—and Mn oxide under identical conditions demonstrate that these two minerals possess equivalent affinity for molybdate (Balistrieri and Chao, 1990). Therefore, we propose that Mn and Fe oxides collectively regulate Mo enrichment in deep-sea sediments.

This conclusion is further substantiated by geochemical evidence from both bulk sample and sequential extraction fractions. The bulk results reveal a significant positive correlation between Fe/Ti and Mo/Ti ratios in both sediment cores, consistent with findings from Pacific Ocean deep-sea sediments (Fig. S1). Similarly, Mn/Ti and Mo/Ti ratios exhibit a significant positive correlation. It is noteworthy that titanium (Ti), a reliable indicator of terrigenous input, was employed for normalization (see

Supplemental Material S1 for details). These relationships are corroborated by sequential extraction experiments, which demonstrate consistent linear correlations among Mn/Ti, Fe/Ti, and Mo/Ti ratios across various geochemical phases (e.g., phosphate-bound, Mn-oxide, and Fe-Mn (oxyhydr)oxide phases; Figure S7). Additionally, results from the sequential phase separation experiment indicate that Fe and Mn play predominant roles in Mo sequestration, despite the relatively large error margins associated with the sequential phase separation extraction technique (including phase overlap errors of approximately 10-30%).

Furthermore, Fe within terrigenous material may contribute to Mo adsorption during early diagenesis, though the extent of this contribution requires validation through sedimentary column simulation experiments.

The corresponding content have been added in the manuscript (Manuscript-marked: Line 432-442). In the revised manuscript, we have changed the descriptions, as: “While Mn oxides have traditionally been regarded as the primary form of Mo in marine oxidizing sediments, our research demonstrates that Fe oxides may also constitute a critical reservoir, a substantial proportion of Mo occurrences (Fig. S3A). In essence, Mn and Fe oxides collectively regulate Mo enrichment in deep-sea sediments. This conclusion is further substantiated by the strong correlations observed between Mo/Ti ratios and both Mn/Ti and Fe/Ti ratios in bulk sediment samples (Fig. 3), relationships that persist across different sequential extraction phases (Fig. S7).”

Reviewer #3 (Remarks to the Author):

Wang et al. presented new Mo isotope data from two cores of deep-sea pelagic sediments, aiming to revise the Mo isotopic fractionations between seawater and its oxic sink, and reconstruct the global marine Mo budgets. This work is very important for the community to understand global marine Mo isotope cycle and use Mo isotope as a paleo-redox proxy, as the pioneer research did not fully solve this issue by only investigating Fe-Mn crust. The manuscript is also well organized and written. However, several issues should be addressed before it can be published. Please see my comments below.

We are very grateful to Reviewer #3 very much for providing insightful and helpful

reviews! They greatly improved the manuscript. Thank you very much!

Major concerns:

1. The Mo isotope signals in bulk marine sediments were reported in the current study, although the sequential extraction of Mo in different phases were also conducted. It is quite important to clarify the Mo phases in bulk sediments, thus it will be helpful if the authors can analyze Mo isotopes in Mn-Fe oxides, besides the bulk sediments. Additionally, the current manuscript didn't discuss the results of sequential extraction of Mo, even though there are only elemental concentration data. I recommend the authors further describe and discuss this part in the revised version.

We appreciate the reviewers' insightful and constructive feedback.

Because the initial sequential extraction phase samples from GC112 were depleted and thus unavailable for molybdenum isotope composition analysis in the Fe-Mn (hydro)oxide fraction, a second sequential extraction (targeting phosphate and Fe-Mn (hydro)oxides) was performed on samples from both the GC112 and XT19 cores. This was done to enable the analysis of Mo isotopes specifically in the Mn-Fe oxide phase. The method, results, and discussion related to this second sequential extraction have been incorporated into the revised manuscript (Lines 516-525, 432-442, and 115-123) and the Supplementary Materials S2 (Lines 67-102).

In the revised manuscript, we have updated the descriptions accordingly.

“To analyze the Mo isotope composition specifically within the Fe-Mn (hydro)oxide fraction, a second sequential extraction targeting both phosphate and Fe-Mn (hydro)oxides was performed on samples from cores GC112 and XT19. The methodology followed established protocols: samples were initially treated with a 1M acetic acid/sodium acetate buffer (pH 4.5) for 24 hours, followed by treatment with a mixed solution of 1M hydroxylamine hydrochloride and 1M HCl at 85°C for 6 hours. Major and trace element analyses were conducted on both extracted fractions, while isotopic measurements were performed exclusively on the Fe-Mn (hydro)oxide phase.

The elemental composition of the residual fraction has not yet been characterized”

“While Mn oxides have traditionally been regarded as the primary form of Mo in marine oxidizing sediments, our research demonstrates that Fe oxides may also constitute a critical reservoir, a substantial proportion of Mo occurrences (Fig. S3A). In essence, Mn and Fe oxides collectively regulate Mo enrichment in deep-sea sediments. This conclusion is further substantiated by the strong correlations observed between Mo/Ti ratios and both Mn/Ti and Fe/Ti ratios in bulk sediment samples (Fig. 3), relationships that persist across different sequential extraction phases (Fig. S7).”

“Mo concentrations, their relative proportions in each extraction phase (derived from sequential extractions), and Mo isotope distributions in the Fe-Mn (hydro)oxides phase are summarized in Tables S2 and S3. The results indicate that Mo is predominantly associated with the Fe-Mn (hydro)oxides phase. Furthermore, the isotopic variation patterns of Mo in this phase for both cores closely mirror those observed in the bulk sediment samples. A detailed evaluation of the sequential extraction methodology and a comprehensive description of the data are available in the Supplementary Materials S2.”

“A Mo elemental mass balance model was utilized to assess the effectiveness of the chemical extractions on the initial sequential extraction phase samples from GC112.

The model is mathematically represented as:

$$[Mo]_{total} = \sum_{i=1}^n [Mo]_i$$

where $[Mo]_i$ denotes the concentration of Mo in phase i . In this analysis, i specifically refers to the phosphate, Mn oxide, Fe (hydro)oxide, and silicate

phases. By accounting for the elemental contributions from each phase, the model provides a comprehensive evaluation of Mo partitioning. The calculated total Mo concentrations ($[Mo]_{total}$) are approximately 20% lower than the measured bulk Mo values (Table S2). Given the strong positive correlation between the two datasets ($R^2=0.95$), it is hypothesized that this discrepancy may result from systematic instrumental errors or other unidentified factors. Major and trace element compositions for the three leachate phases are presented in Table S3. For the second batch of Mo sequential extraction experiments on core sediments from GC112 and XT19, equilibrium calculations were not performed due to the lack of residual phase data. Nevertheless, the combined Mo content in the extracted phases constitutes over 80% of the measured bulk Mo values. Isotopic mass balance calculations were not performed because Mo isotope data could not be obtained for several phases, as their Mo concentrations were too low to allow $\delta^{98}Mo$ measurement.

In the initial sequential extraction phase of the GC112 samples, Mo concentrations relative to the original sample weight were as follows: 0.7–3.5 $\mu g g^{-1}$ in the phosphate phase (1.0%–10.3% of bulk sample Mo), 2.2–42.4 $\mu g g^{-1}$ in the Mn oxide phase (2.4%–34.7% of bulk sample Mo), 7.2–47.3 $\mu g g^{-1}$ in the Fe oxide phase (29.5%–51.4% of bulk sample Mo), and 2.5–18.2 $\mu g g^{-1}$ in the residue phase (7.4%–19.9% of bulk sample Mo) (Table S3). For the second batch of Mo sequential extraction experiments, results varied by core (Table S4). In the GC112 core sediments, Mo concentrations were 3.8–21.4 $\mu g g^{-1}$ in the phosphate phase (3.7%–34.4% of bulk sample Mo) and 7.6–70.9 $\mu g g^{-1}$ in the Fe-Mn oxide phase (44.2%–61.5% of bulk sample Mo). In the XT19 core

sediments, Mo concentrations measured 0.96–1.45 $\mu\text{g g}^{-1}$ in the phosphate phase (2.82%–5.49% of bulk sample Mo) and 18.0–36.2 $\mu\text{g g}^{-1}$ in the Fe-Mn oxide phase (84.6%–92.4% of bulk sample Mo). Regarding isotopic composition, the $\delta^{98}\text{Mo}$ values in the Fe-Mn (hydro)oxide phase of the GC112 samples ranged from -0.86‰ to 0.04‰ (Table S4), indicating a slight depletion compared to bulk rock values. Conversely, the $\delta^{98}\text{Mo}$ signatures in the Fe-Mn (hydro)oxide phase of XT19 samples (from -0.47‰ to 0.04‰) closely approximated those of their corresponding bulk samples. Notably, despite these minor differences, the isotopic variation patterns in the Fe-Mn (hydro)oxide phase of both cores broadly mirrored those exhibited by the bulk samples.”

2. Changes in scavenging patterns of Mo within the sediment piles. The key finding in this study is the increased $\delta^{98}\text{Mo}$ values with water depth, positive correlations between $\delta^{98}\text{Mo}$ and Mo/Al, and negative correlations between $\delta^{98}\text{Mo}$ and Mn/Mo ratios. However, this observation notably supports that the Mo precipitation in the deeper sediment pile may reflect the authigenic Mn accumulation under more reducing conditions. Additionally, some reducing sediments can also have high Mn contents (e.g., Gullmar Fjord, Sweden) (Goldberg et al., 2012 Chemical Geology). Thus, one critical point is to identify the types of Mn burial and redox conditions of pore water within the sediments, which should be fully constrained in this study. Any pore water data or redox-sensitive elemental analysis (e.g., Mo-U-V) should help to clarify this point.

We appreciate the reviewers' insightful and constructive feedback. As pore water was not collected during these voyages (Zhang et al., 2024; Bai et al., 2025), the redox state of the studied core was inferred using geochemical indicators from pore water in cores retrieved from adjacent regions.

In the Western Pacific study area, bottom waters exhibit relatively high dissolved oxygen concentrations (Hondt et al., 2015; Chen et al., 2022). Pore water profiles throughout the core reveal nitrate and sulfate concentrations comparable to ambient seawater levels, while dissolved Fe and Mn concentrations remain very low (0.05–0.3

$\mu\text{mol/L}$) (Deng, 2021). These low concentrations contrast sharply with the significantly higher Fe (25–200 $\mu\text{mol/L}$) and Mn (16–180 $\mu\text{mol/L}$) levels characteristic of pore waters under nearshore reducing conditions (Goldberg et al., 2012). Collectively, this evidence suggests predominantly oxic conditions within the sediment core. Such conditions are consistent with previous studies demonstrating that modern bottom-water oxygen concentrations in deep-sea environments (typically >4,000 m depth) exceed 140 $\mu\text{mol/kg}$ (Garcia et al., 2019), and that oxygen may plausibly penetrate to the volcanic basement (Hans et al., 2012; Hondt et al., 2004, 2015).

The corresponding revisions have been incorporated into the manuscript (Lines 153–213).

“This observed pattern of increasing Mn/Ti and Fe/Ti ratios with depth aligns with previous findings in numerous deep-sea sediment cores^{6, 13, 15, 19, 20, 21, 22, 26, 27}. Two primary mechanisms have been proposed for this Mn and Fe enrichment: (1) reductive dissolution and subsequent re-oxidation, where Mn and Fe oxide reduction in pore water generates dissolved Mn and Fe ions that then re-oxidize and accumulate as oxides in deeper sediment layers²⁶; and (2) oxidative precipitation, involving direct Fe and Mn precipitation from the water column under oxidizing bottom-water conditions during sediment deposition. The latter mechanism is governed by the availability of Fe and Mn in bottom waters and the sediment accumulation rate.

Bottom waters in the Western Pacific study area exhibit high dissolved oxygen concentrations^{16, 28}. Pore water profiles from cores in adjacent regions show SO_4^{2-} and NO_3^- concentrations equal to or exceeding those of seawater. Dissolved Fe and Mn concentrations are low (0.05–0.3 $\mu\text{mol/L}$)²⁹, contrasting sharply with the significantly higher Fe (25–200 $\mu\text{mol/L}$) and Mn (16–180 $\mu\text{mol/L}$) levels typical of pore waters in

nearshore reducing environments³⁰. These values indicate that predominantly oxic conditions prevail within the sediment column. These conditions align with previous research indicating that modern bottom-water oxygen concentrations in deep-sea environments (typically >4,000 m depth) exceed 140 $\mu\text{mol/kg}$ ³¹, and that oxygen can penetrate to the volcanic basement^{28, 32, 33}. Consequently, enhanced precipitation under oxidizing bottom-water conditions represents the most plausible explanation for the observed Fe and Mn enrichment with depth. Notably, while the dissolved Fe and Mn concentrations in pore water from adjacent regions (0.05–0.3 $\mu\text{mol/L}$) are low compared to those in reducing environments, they exceed typical seawater levels. Research suggests that, under oxic conditions, such elevated dissolved Fe primarily originates from colloidal Fe released during the non-reductive diagenesis of tephra sediments^{34, 35, 36}. By analogy, the observed Mn enrichment may also largely result from the release of colloidal Mn via similar non-reductive diagenetic processes involving tephra material.

The Mo/Ti ratios in both sediment cores exhibit variation patterns similar to those of Mn/Ti and Fe/Ti (Fig. 2), suggesting that sedimentary Mo might co-precipitate with Fe and Mn oxides. However, if Mo were to deposit and enrich concurrently with Fe and Mn oxides, the Mn/Mo and Fe/Mo ratios in the sediment, along with its Mo isotopic composition, should align with values from the top surface of both sediment cores and remain constant with depth. In contrast, our data show that Mn/Mo and Fe/Mo ratios gradually decrease with depth, while Mo isotopes become progressively heavier. Consequently, we hypothesize that post-sedimentation, sedimentary Mo is likely

influenced by early diagenetic processes, such as the infiltration and addition of Mo from bottom waters. This hypothesis aligns with previous findings that early diagenetic processes are primary factors influencing the fractionation of rare earth elements and Zn isotope composition in Pacific Ocean deep-sea sediments^{23, 29}. This is further corroborated by the observation that dissolved Mo concentrations in the pore waters of Western Pacific sediments are generally lower than bottom water concentrations and decrease with depth (from approximately 110 nM—the seawater Mo concentration—at the sediment-water interface to about 20 nM at 15 meters below seafloor (mbsf)³⁷). This indicates continuous Mo transport from bottom water into the sediment via diffusion and subsequent adsorption onto Fe and Mn oxides. Due to higher Mn and Fe content and/or prolonged adsorption in deeper layers, these sediments exhibit greater Mo enrichment than shallower layers. This leads to enhanced Mo accumulation at depth (Fig. 2C and Fig. S2A), and Mo/Ti ratios show a positive correlation with Mn/Ti and Fe/Ti ratios (Fig. 3A and Fig. 1B). Conversely, Fe and Mn concentrations in pore water are slightly higher than in bottom water, which precludes their sustained transport from bottom water into sediments. Ultimately, these processes lead to a gradual decrease in the Mn/Mo and Fe/Mo ratios with depth^{19, 20, 21}.”

3. The effects of isotopic fractionations from deeper sediments on modern global seawater Mo mass balance. The residence time of Mo in modern ocean is around 400 kyr. If the deeper part of the sediment profile in this study was deposited much earlier than 400 ka, the variations of their Mo isotope compositions may not influence modern seawater Mo budgets. It will be helpful to present the depositional age or relative age model of the studied sediment profiles.

We sincerely appreciate the reviewers' insightful and constructive feedback.

Deep-sea sediments are integral to the modern oceanic geochemical cycle, serving as crucial reservoirs that influence the compositional cycling of elements and isotopes through exchanges with seawater, including processes like early diagenesis. The ages of the sediment cores have been determined: the XT19 core sample is approximately 2 Ma, and the GC112 sample is estimated to be around 10 Ma, based on sedimentation rates (Yi et al., 2021; Zhang et al., 2024). Although the ages of the XT19 and GC112 sediment cores exceed the residence time of Mo in the ocean (0.44 Ma), their interactions with seawater—primarily driven by early diagenetic processes—occur on significantly shorter timescales. Previous studies have shown that the infiltration rate of Pacific Ocean bottom water into sediments ranges from 1.2 to 10 cm/year (Hesse and Schacht, 2011; Cheng et al., 2024). Using this range, we estimate that the timescales for seawater-sediment exchange for XT19 and GC112 are on the order of tens to hundreds of years. This duration is considerably shorter than both the Mo residence time and the oceanic mixing time, allowing for substantial alterations to the geochemical composition of the porewater, which in turn interacts with the sediments. These interactions can influence the sedimentary record of the modern ocean Mo cycle within its residence time. Furthermore, the Mo isotope compositions in these cores have undergone only minor changes since their deposition approximately 10 Ma ago (Fig. S5), a finding consistent with the temporal pattern of oceanic Mo isotope variation reconstructed from Fe-Mn nodules. Collectively, these observations suggest that the Mo isotopic signatures documented in the deep-sea sediments cannot be attributed to fluctuations in the isotopic composition of deep seawater since the Miocene (Fig. S5). Therefore, the average $\delta^{98}\text{Mo}$ value ($-0.09 \pm 0.23\text{‰}$) derived from these deep-sea sediments provides a reliable proxy for the oxic sink value in the modern ocean (Fig. S5). Similar methodologies have been employed in prior research, where deep-sea sediments were identified as important sinks for other isotopes such as zinc (Zn) and copper (Cu) (Zhang et al., 2024; Little et al., 2025).

The corresponding revisions have been incorporated into the manuscript (Lines 350-366). In the revised manuscript, we have changed the descriptions as follows:

“Deep-sea sediments, as integral components of the modern ocean system, play a vital role in regulating marine geochemical cycles through early diagenetic processes. Previous research indicates that seawater infiltration rates—a key driver of early diagenesis in deep-sea settings—are relatively high, ranging from 1.2 to 10 cm/year⁴⁹,⁵⁰. Based on the core depths analyzed in this study, the estimated timescale for seawater infiltration spans several decades to several hundred years. This duration is substantially shorter than the oceanic Mo residence time of 0.44 Ma¹². Furthermore, the global oceanic Mo isotopic composition, reconstructed from Fe-Mn nodules since the Cenozoic era, has remained stable¹. Consistent with this stability, Mo isotope compositions in the studied cores have undergone only minor changes since their deposition approximately 10 million years ago (Fig. S5). Collectively, these observations suggest that the exchange of Mo isotopes between deep-sea sediments and the ocean has largely been in a stable equilibrium state since the time of sediment deposition. Therefore, the average $\delta^{98}\text{Mo}$ value ($-0.09 \pm 0.23\text{‰}$) derived from these deep-sea sediments provides a reliable proxy for the oxic sink value in the modern ocean.”

References

- Cheng, Y., Cai, P., Chen, H., Yuan, L., Jiang, X., Zhang, S., Chen, Y., Luo, Y., and Sohrin, Y., 2024, Nitrate and silicate fluxes at the sediment–water interface of the deep North Pacific Ocean illuminated by $^{226}\text{Ra}/^{230}\text{Th}$ disequilibria: *Geochimica et Cosmochimica Acta*, v. 383, p. 81-91.
- Hesse, R., and Schacht, U., 2011, Chapter 9 - Early Diagenesis of Deep-Sea Sediments, in HüNeke, H., and Mulder, T., eds., *Developments in Sedimentology*, Volume 63, Elsevier, p. 557-713.
- Little, S. H., de Souza, G. F., and Xie, R. C., 2025, Metal stable isotopes in the marine realm, in Anbar,

A., and Weis, D., eds., *Treatise on Geochemistry*: Oxford, Elsevier, p. 285-332.

Zhang, G. L., Zhu, Y. T., Deng, Y. N., Cao, J., Wang, P. C., Yang, A., He, G. W., Zhao, B., and Zhao, M.

Y., 2024, Balancing the oceanic Zn isotope budget: The key role of deep-sea pelagic sediments:

Geology, v. 52, no. 10, p. 789-793.

Specific items:

Line 34. Better to say “proxy for ancient oceanic oxygen levels”.

It has been changed (Lines 35).

Line 58. The reference 7 (Gill et al., 2011 Nature) didn't mention Mo proxy. It's better to cite Wei et al. (2021) *Earth-Science Reviews*, instead.

We have replaced Gill et al. (2011) with Wei et al. (2021) (Lines 59).

Line 112-114. Why can low Fe and Mn concentrations of the sediments denote limited hydrothermal influence? I think the dilution of detrital materials (Si, Al) should be considered. Thus, the ratios of Fe/Al, Mn/Al can be used to trace the source of oxides in these sediments.

We have replaced Fe and Mn concentrations of the sediments with the ratios of Fe/Ti, Mn/Ti (Lines 137-142).

Line 130-131. Progressive Mn oxidation in pore waters should lead to the decreases in dissolved Mn through the sediment pile. In this view, the deeper sediment layers should have lower Mn accumulations. Does high Mn concentration of deeper sediments represent the accumulation of Mn oxides, or other Mn-bearing minerals (rhodochrosite)?

Thank you for your insightful feedback. A high Mn concentration in deeper sediments appears to be a common phenomenon, as demonstrated by the previous studies (Bi et al., 2021; Tanaka et al., 2020, 2023; Ren et al., 2021). This occurrence may be closely associated with the depositional oxic environment.

The corresponding content have been added in the manuscript (Line 153-180).

In the revised manuscript, we have changed the descriptions, as: “This observed pattern of increasing Mn/Ti and Fe/Ti ratios with depth aligns with previous findings in numerous deep-sea sediment cores^{6, 13, 15, 19, 20, 21, 22, 26, 27}. Two primary mechanisms have been proposed for this Mn and Fe enrichment: (1) reductive dissolution and subsequent re-oxidation, where Mn and Fe oxide reduction in pore water generates dissolved Mn

and Fe ions that then re-oxidize and accumulate as oxides in deeper sediment layers²⁶; and (2) oxidative precipitation, involving direct Fe and Mn precipitation from the water column under oxidizing bottom-water conditions during sediment deposition. The latter mechanism is governed by the availability of Fe and Mn in bottom waters and the sediment accumulation rate.

Bottom waters in the Western Pacific study area exhibit high dissolved oxygen concentrations^{16, 28}. Pore water profiles from cores in adjacent regions show SO_4^{2-} and NO_3^- concentrations equal to or exceeding those of seawater. Dissolved Fe and Mn concentrations are low (0.05–0.3 $\mu\text{mol/L}$)²⁹, contrasting sharply with the significantly higher Fe (25–200 $\mu\text{mol/L}$) and Mn (16–180 $\mu\text{mol/L}$) levels typical of pore waters in nearshore reducing environments³⁰. These values indicate that predominantly oxic conditions prevail within the sediment column. These conditions align with previous research indicating that modern bottom-water oxygen concentrations in deep-sea environments (typically >4,000 m depth) exceed 140 $\mu\text{mol/kg}$ ³¹, and that oxygen can penetrate to the volcanic basement^{28, 32, 33}. Consequently, enhanced precipitation under oxidizing bottom-water conditions represents the most plausible explanation for the observed Fe and Mn enrichment with depth.”

Line 148-150. Do lower Mn/Mo and Fe/Mo ratios of deeper sediments reflect the effects of local pore water redox condition on Mo scavenging (i.e., the transition from oxide absorption to reducing burial), as the precipitation rates of Mo under reducing conditions are much higher than those of oxide absorption?

This issue is similar to the previous point (Major 2).

In the Western Pacific study area, bottom waters contain relatively high concentrations of dissolved oxygen (Hondt et al., 2015; Chen et al., 2022). Pore water profiles across the core indicate nitrate and sulfate levels comparable to those of ambient seawater, while dissolved Fe and Mn concentrations remain notably low (0.05–0.3 $\mu\text{mol/L}$; Deng, 2021). These values stand in stark contrast to the substantially higher iron (25–200 $\mu\text{mol/L}$) and manganese (16–180 $\mu\text{mol/L}$) concentrations typically observed in pore waters from nearshore reducing environments (Goldberg et al., 2012). Collectively,

these findings indicate that the sediment core is predominantly oxic. This interpretation aligns with previous studies showing that bottom-water oxygen concentrations in deep-sea settings (generally exceeding 4,000 m in depth) are typically above 140 $\mu\text{mol/kg}$ (Garcia et al., 2019), and that oxygen may plausibly penetrate to the volcanic basement (Hans et al., 2012; Hondt et al., 2004, 2015).

The corresponding revisions have been incorporated into the manuscript (Lines 153-213).

Line 222-225. Are all these Mo concentration data derived from highly oxic Mo sink (i.e., absorbed by Fe-Mn oxides)?

Thank you for your insightful feedback. Previous studies demonstrating that modern bottom-water oxygen concentrations in deep-sea environments (typically >4,000 m depth) exceed 140 $\mu\text{mol/kg}$ (Garcia et al., 2019), and that oxygen may plausibly penetrate to the volcanic basement (Hans et al., 2012; Hondt et al., 2004, 2015).

The corresponding revisions have been incorporated into the manuscript (Lines 153-213).

References

- Garcia, H. E., Weathers, K., Paver, C. R., Smolyar, I., Boyer, T. P., Locarnini, M. M., et al. (2019). World Ocean Atlas 2018, Volume 3: Dissolved Oxygen, Apparent Oxygen Utilization, and Dissolved Oxygen Saturation. Retrieved from <https://archimer.ifremer.fr/doc/00651/76337/>
- Garcia H. E., K.W. Weathers, C.R. Paver, I. Smolyar, T.P. Boyer, R.A. Locarnini, M.M. Zweng, A.V. Mishonov, O.K. Baranova, D. Seidov, and J.R. Reagan (2019). World Ocean Atlas 2018, Volume 3: Dissolved Oxygen, Apparent Oxygen Utilization, and Dissolved Oxygen Saturation. A. Mishonov Technical Editor. NOAA Atlas NESDIS 83, 38pp.
- Hans Røy et al., Aerobic Microbial Respiration in 86-Million-Year-Old Deep-Sea Red Clay. *Science* 336,922-925(2012). DOI:10.1126/science.1219424.
- Steven D'Hondt et al., Distributions of Microbial Activities in Deep Subseafloor Sediments. *Science* 306,2216-2221(2004). DOI:10.1126/science.1101155.
- D'Hondt, S., Inagaki, F., Zarikian, C. et al. Presence of oxygen and aerobic communities from sea floor to basement in deep-sea sediments. *Nature Geosci* 8, 299–304 (2015). <https://doi.org/10.1038/ngeo2387>.

Line 253. Is high temperature hydrothermal fluid a Mo sink?

Thank you for your insightful feedback. Miller et al. (2011) proposed that high-temperature hydrothermal vent fluids are relatively insignificant sinks compared to their riverine sources, contributing only 0.4% (in terms of molybdenum).

Fig. 1. Mo contents of GC112 core are quite high (up to 120 ppm). This value is generally from reducing or anoxic sediments as the authors also compiled here.

Thank you for your insightful feedback. This issue is similar to the previous point (Major 2).

Geochemical indicators in the pore water of Pacific sediments demonstrate that these sediment cores predominantly experience oxic conditions. Consequently, the Mo enrichment observed in the GC112 core cannot be ascribed to reducing or anoxic environments. In contrast, previous research has shown that molybdenum (Mo) concentrations are frequently elevated in oxic deep-sea sediments. For example, Ahmad et al. (2021) and Yu et al. (2021) reported Mo levels exceeding 100 ppm in deep-sea sediments, which were strongly associated with elevated iron and manganese concentrations.

The corresponding revisions have been integrated into the manuscript (Lines 153–213).

References

- Ahmad Q, et al. The Molybdenum isotope subduction recycling conundrum: A case study from the Tongan subduction zone, Western Alps and Alpine Corsica. *Chem Geol* 576, 120231 (2021).
- Yu M, et al. The transfer of rare earth elements during early diagenesis in REY-rich sediments: An example from the Central Indian Ocean Basin. *Ore Geol Rev* 136, 104269 (2021).

Reviewer #1 (Remarks to the Author):

Congratulations on this important contribution. I am fully satisfied with the revisions and see no obstacle to publication.

Reviewer #2 (Remarks to the Author):

Review of the revised version of the manuscript NCOMMS-25-10252A “Revised Oceanic Mo Isotope Budget from Deep-Sea Pelagic Sediments” by Wang et al. submitted to Nature Communications.

Similar to my statement in the first review of this manuscript and based on the presented data, I think the proposed scenario of Mo transport from bottom water into the sediment via diffusion and subsequent adsorption onto Fe and Mn (hydro)oxides is a possible scenario. However concerning the here presented data two major concerns stay:

Answer: We sincerely thank Reviewer #2 for their constructive feedback and careful evaluation of our revised manuscript. The comments have helped us to further clarify our interpretations and strengthen our arguments. We agree that the scenario of Mo transport via diffusion and subsequent adsorption is complex, and we appreciate the opportunity to address the remaining concerns regarding the specific roles of Fe and Mn (hydro)oxides and the interpretation of geochemical proxies.

1) The influence on Fe oxides on Mo and higher Mo association within the Fe (hydr)oxide leach fractions

Answer: We appreciate the reviewer's critical assessment regarding the evidence for molybdenum's (Mo) association with iron (Fe) (hydro)oxides. While we concur that manganese (Mn) oxides are the principal host for Mo in the studied sediments, the contribution of Fe oxides cannot be disregarded. Currently, there is no conclusive evidence to suggest that Fe is uninvolved in the Mo cycle. On the contrary, our data reveal a positive correlation between the Mo/Ti ratio and both the Mn/Ti and Fe/Ti

ratios across various phases and the bulk sample in GC112 and XT19 core. This finding indicates that Fe contributes significantly to the Mo cycle. Therefore, we maintain that while Mo is primarily hosted in manganese oxides, iron oxides serve as a secondary host. This underscores the significant geochemical linkage between Mn, Fe, and Mo enrichment, leading us to propose that Mn and Fe oxides collectively regulate Mo enrichment in deep-sea sediments.

Here three observations from the authors data argue against this significant Fe influence of Mo:

a) If authigenic Fe input has a significant affect the Mo removal, why in Mo/Ti values between two cores behave similar to Mn/Ti ratios and not Fe/Ti (Fig. 2)? In core XT19 higher Fe/Ti values compared to core GC112 would have an impact in authigenic Mo enrichment (Mo/Ti). This is not the case.

Answer: We appreciate the reviewer's insightful comments. Regarding the observation that the Fe/Ti ratio of the XT19 core is slightly higher than that of GC112, it is important to emphasize that this ratio in deep-sea sediments is influenced by both authigenic iron and, more significantly, variations in terrigenous (detrital) material. More than 80% of iron in these environments originates from terrestrial sources like atmospheric dust and volcanic ash (Olivarez et al., 1991; Dunlea et al., 2017; 2021; Yi et al., 2021; Yu et al., 2021; Zhang et al., 2024; Tegler et al., 2025). Although it is often assumed that the Fe/Ti ratio of terrigenous material is relatively stable, these inputs are inherently heterogeneous. They exhibit spatial and compositional variability due to differences in source material and alterations during transport (Guo et al., 2004; Mahowald et al., 2005; O'Day et al., 2022; Wang et al., 2022). Consequently, even minor variations in terrigenous inputs can lead to notable fluctuations in the Fe/Ti ratios of deep-sea sediments.

In this study, the two sediment cores, XT19 and GC112, were sourced from locations with distinct provenances and atmospheric transport pathways. Specifically, core GC112 is influenced by both atmospheric particulates and volcanic ash, whereas the

dust contributing to core XT19 has undergone more extensive long-range transport (Yi et al., 2021; Zhang et al., 2024). These combined factors result in the minor variations observed in the Fe/Ti ratios between the two cores.

Despite these small differences, the variations are not statistically significant on a global scale, and the Fe/Ti versus Mo/Ti data from both cores align with the established global trend (Fig. S1). Therefore, the observed discrepancies in Fe/Ti ratios are insufficient to support the conclusion that "iron plays no role whatsoever in regulating the molybdenum geochemical cycle."

References

- Olivarez, A. M., Owen, R. M., and Rea, D. K., 1991, Geochemistry of eolian dust in Pacific pelagic sediments: Implications for paleoclimatic interpretations: *Geochimica et Cosmochimica Acta*, v. 55, no. 8, p. 2147-2158.
- Yu M, et al. The transfer of rare earth elements during early diagenesis in REY-rich sediments: An example from the Central Indian Ocean Basin. *Ore Geol Rev* 136, 104269 (2021).
- Dunlea, A. G., Tegler, L. A., Peucker-Ehrenbrink, B., Anbar, A. D., Romaniello, S. J., and Horner, T. J., 2021, Pelagic clays as archives of marine iron isotope chemistry: *Chemical Geology*, v. 575, p. 120201.
- Dunlea, A. G., Murray, R. W., Santiago Ramos, D. P., and Higgins, J. A., 2017, Cenozoic global cooling and increased seawater Mg/Ca via reduced reverse weathering: *Nature Communications*, v. 8, no. 1, p. 844.
- Yi L, et al. Magnetostratigraphy of Abyssal Deposits in the Central Philippine Sea and Regional Sedimentary Dynamics During the Quaternary. *Paleoceanogr Paleocl* 37, e2021PA004365 (2022).
- Zhang GL, et al. Balancing the oceanic Zn isotope budget: The key role of deep-sea pelagic sediments. *Geology* 52, 789-793 (2024).
- Guo, Z. G., Feng, J. L., Fang, M., Chen, H. Y., and Lau, K. H., 2004, The elemental and organic characteristics of PM2.5 in Asian dust episodes in Qingdao, China, 2002: *Atmospheric Environment*, v. 38, no. 6, p. 909-919.
- Mahowald, N. M., Baker, A. R., Bergametti, G., Brooks, N., Duce, R. A., Jickells, T. D., Kubilay,

- N., Prospero, J. M., and Tegen, I., 2005, Atmospheric global dust cycle and iron inputs to the ocean, v. *Global Biogeochemical Cycles* 19, no. 4.
- O'Day, P. A., Pattammattel, A., Aronstein, P., Leppert, V. J., and Forman, H. J., 2022, Iron Speciation in Respirable Particulate Matter and Implications for Human Health: *Environ Sci Technol*, v. 56, no. 11, p. 7006-7016.
- Wang, Y., Wu, L., Hu, W., Li, W., Shi, Z., Harrison, R. M., and Fu, P., 2022, Stable iron isotopic composition of atmospheric aerosols: An overview: *npj Climate and Atmospheric Science*, v. 5, no. 1, p. 75.
- Ali, J., Tuzen, M., Shaikh, Q.-u.-a., Jatoi, W. B., Feng, X., Sun, G., and Saleh, T. A., 2024, A review of sequential extraction methods for fractionation analysis of toxic metals in solid environmental matrices: *TrAC Trends in Analytical Chemistry*, v. 173, p. 117639.
- Sutherland, R. A., 2010, BCR®-701: A review of 10-years of sequential extraction analyses: *Analytica Chimica Acta*, v. 680, no. 1, p. 10-20.
- Tegler, L. A., Horner, T. J., Nielsen, S. G., Heard, A. W., Squires, K. R., Severmann, S., Peucker-Ehrenbrink, B., Blusztajn, J., and Dunlea, A. G., 2025, Evolution of the South Pacific's Iron Cycle Over the Cenozoic, v. 40, no. 7, p. e2025PA005149.

In the different leached fraction of MnO and Fe (hydro) oxide phases, Mo concertation in both leach fractions covary with Mn concertation and while a negative covariation is observed with Fe! Does this show that Mo is likely bound to Mn phases?

Answer: With respect to the various leached fractions of MnO and Fe (hydro)oxide phases, this issue is analogous to the question raised in part (c).

b) As stated in my last review, Co concertation covary with Mn and Mo concentrations within the sediments presented here suggesting Mn as primary Mo host as for Co

Answer: We appreciate the reviewer's thoughtful feedback and recognize the importance of clarifying the respective roles of manganese and iron oxides in regulating molybdenum (Mo) enrichment. We concur that Mo is primarily associated with

manganese oxides in deep-sea sediments, a conclusion supported by the strong positive covariation between Mo and Mn concentrations. The significant correlation between Co and Mn further highlights the dominance of Mn oxides as primary Mo hosts.

However, our sequential extraction results demonstrate that a measurable proportion of Mo is consistently recovered in the Fe (hydro)oxide extraction phase (see Table S2), indicating that a fraction of sedimentary Mo is indeed linked to Fe oxides. While we acknowledge that the methodological limitations of sequential extraction can introduce quantitative uncertainty due to potential overlap between phases (Elaborate specifically in in part (c)), this finding is robust. Also, our data reveal a positive correlation between the Mo/Ti ratio and Fe/Ti ratios across various phases and the bulk sample in GC112 and XT19 cores (Fig.3 and Fig.S7). This finding indicates that Fe contributes significantly to the Mo cycle. This observation is supported by previous research showing that MoO_4^{2-} can adsorb onto Fe-(oxy)hydroxide surfaces under certain geochemical conditions, particularly in Fe-rich microenvironments or during early diagenesis (e.g., Goldberg et al., 2009; Wasylenzi et al., 2011). Notably, Mo adsorption experiments conducted under identical conditions with both ferrihydrite—a key component comprising 50–70% of the total Fe-(oxy)hydroxide pool in most marine environments—and Mn oxide have demonstrated that these two minerals possess an equivalent affinity for molybdate (Balistrieri and Chao, 1990).

Furthermore, Mn and Fe oxides commonly co-precipitate or exist in close spatial association within marine sediments. This proximity suggests that while Mn oxides serve as the primary host, some Mo may be retained through subsequent adsorption onto Fe oxides, potentially following the dissolution of Mn phases or via surface complexation on adjacent Fe-rich particles. This coupled pathway is mechanistically plausible and aligns with our empirical data, which reveal positive correlations between Mo/Ti and both Mn/Ti and, to a lesser extent, Fe/Ti.

In summary, although Mn oxides are the principal host for Mo, Fe oxides also contribute discernibly to its retention in deep-sea sediments.

c) Plotting the different Mo, Mn and Fe concentration data from the different leaching

steps (Table S2), there is a covariation between Mn and Mo concentrations within the Mn oxide phase and eFe (hydro) oxide phase sequential extraction. Although Mn concentration are quite low in second leach step, does this show that during both leaching steps the Mo is dominantly released from Mn-phases or that these different leaching steps were not quantitatively able to fully discriminate between both phases?

Answer: We sincerely appreciate the reviewer's valuable comments and welcome the opportunity to clarify our interpretation of the sequential extraction data.

Regarding the suggestion to plot absolute concentrations, we maintain that this approach is misleading, particularly for iron. In these deep-sea sediments, Fe is overwhelmingly terrigenous in origin (>80%), with major contributions from atmospheric dust and volcanic ash (Olivarez et al., 1991; Dunlea et al., 2021; Yi et al., 2021). These terrestrial inputs are rich in iron oxides that are susceptible to dissolution during extraction, causing absolute Fe concentrations in the leachates to substantially overestimate the authigenic fraction and obscure its true geochemical behavior (Guo et al., 2004; Mahowald et al., 2005).

To isolate the authigenic signal, normalization against a conservative lithogenic element like Ti is essential. This method corrects for the dominant influence of terrigenous Fe, allowing the relationship between authigenic Mo and Fe to be accurately assessed. The negative Mo-Fe correlation observed in the raw concentrations (Fig. S3) is an artifact of this uncorrected terrestrial input. In contrast, the positive correlations between Mo/Ti and Fe/Ti ratios across various extraction phases (Fig. S7) align with established global trends and support a genuine association for authigenic Fe in Mo sequestration. This demonstrates that the apparent negative trend in the raw data does not negate the involvement of the authigenic Fe phase. Conversely, because Mn and Mo are primarily authigenic with minimal terrestrial input, their concentration profiles are not significantly altered by normalization.

We concur with the reviewer's observation from our extraction data (Fig. S3) that the measured Mo content appears higher in the Fe-associated phase than in the Mn-associated phase. However, we attribute this to the inherent limitations of the sequential

extraction method itself. This technique is semi-quantitative, and significant issues such as phase overlap and inter-step elemental migration are well-documented (Ali et al., 2024; Sutherland et al., 2010). For instance, Mo liberated during the dissolution of Mn oxides can be reabsorbed onto remaining solid phases (including Fe oxides), leading to its overestimation in subsequently targeted fractions. Consequently, the apparent enrichment of Mo in the Fe-oxide phase should be viewed as a methodological artifact, not as direct evidence that Fe oxides are the primary host for Mo in the natural system. We have added a statement to the revised manuscript to explicitly acknowledge this uncertainty (Line 484-489).

Despite these methodological caveats, our collective findings contradict the conclusion that Fe is uninvolved in the Mo cycle. Multiple lines of evidence—including the positive correlations between Mo/Ti, Mn/Ti, and Fe/Ti in bulk samples and extracted phases in GC112 and XT19 cores, alongside the alignment of our data with global trends—indicate that Fe oxides act as a secondary host for Mo. While the strong covariation with Co validates the primary role of Mn oxides, this does not preclude a contribution from Fe. In summary, we maintain that while Mo is predominantly hosted in manganese oxides, iron oxides also play a discernible role.

Relevant content has been added in the revised manuscript (Line 484-489 and 770-774), as:

“It is important to note that this method has inherent limitations. A substantial portion of Fe and Mo remained in the residual fraction, suggesting that the extraction of crystalline Fe (hydro)oxides in step 5 may have been incomplete. Furthermore, significant uncertainty exists in quantifying Mo within specific phases due to the potential for partial dissolution and elemental redistribution during the extraction process, as the reagents are operationally defined and not perfectly selective for individual mineral hosts.”

“Fe in the deep-sea sediments of cores XT19 and GC112, which is present predominantly as (hydro)oxides, originates mainly from atmospheric dust and volcanic ash. Consequently, normalizing Fe concentrations to Ti across the various extraction phases is essential for isolating the authigenic Fe signal.”

The relevant content pertaining to iron is changed in the revised manuscript (Lines 400–417).

References

- Olivarez, A. M., Owen, R. M., and Rea, D. K., 1991, Geochemistry of eolian dust in Pacific pelagic sediments: Implications for paleoclimatic interpretations: *Geochimica et Cosmochimica Acta*, v. 55, no. 8, p. 2147-2158.
- Yu M, et al. The transfer of rare earth elements during early diagenesis in REY-rich sediments: An example from the Central Indian Ocean Basin. *Ore Geol Rev* 136, 104269 (2021).
- Dunlea, A. G., Tegler, L. A., Peucker-Ehrenbrink, B., Anbar, A. D., Romaniello, S. J., and Horner, T. J., 2021, Pelagic clays as archives of marine iron isotope chemistry: *Chemical Geology*, v. 575, p. 120201.
- Yi L, et al. Magnetostratigraphy of Abyssal Deposits in the Central Philippine Sea and Regional Sedimentary Dynamics During the Quaternary. *Paleoceanogr Paleocl* 37, e2021PA004365 (2022).
- Zhang GL, et al. Balancing the oceanic Zn isotope budget: The key role of deep-sea pelagic sediments. *Geology* 52, 789-793 (2024).
- Guo, Z. G., Feng, J. L., Fang, M., Chen, H. Y., and Lau, K. H., 2004, The elemental and organic characteristics of PM_{2.5} in Asian dust episodes in Qingdao, China, 2002: *Atmospheric Environment*, v. 38, no. 6, p. 909-919.
- Mahowald, N. M., Baker, A. R., Bergametti, G., Brooks, N., Duce, R. A., Jickells, T. D., Kubilay, N., Prospero, J. M., and Tegen, I., 2005, Atmospheric global dust cycle and iron inputs to the ocean, v. 19, no. 4.
- O'Day, P. A., Pattammattel, A., Aronstein, P., Leppert, V. J., and Forman, H. J., 2022, Iron Speciation in Respirable Particulate Matter and Implications for Human Health: *Environ Sci Technol*, v. 56, no. 11, p. 7006-7016.
- Wang, Y., Wu, L., Hu, W., Li, W., Shi, Z., Harrison, R. M., and Fu, P., 2022, Stable iron isotopic composition of atmospheric aerosols: An overview: *npj Climate and Atmospheric Science*, v. 5, no. 1, p. 75.
- Ali, J., Tuzen, M., Shaikh, Q.-u.-a., Jatoi, W. B., Feng, X., Sun, G., and Saleh, T. A., 2024, A review

of sequential extraction methods for fractionation analysis of toxic metals in solid environmental matrices: *TrAC Trends in Analytical Chemistry*, v. 173, p. 117639.

Sutherland, R. A., 2010, BCR®-701: A review of 10-years of sequential extraction analyses: *Analytica Chimica Acta*, v. 680, no. 1, p. 10-20.

Tegler, L. A., Horner, T. J., Nielsen, S. G., Heard, A. W., Squires, K. R., Severmann, S., Peucker-Ehrenbrink, B., Blusztajn, J., and Dunlea, A. G., 2025, Evolution of the South Pacific's Iron Cycle Over the Cenozoic, v. 40, no. 7, p. e2025PA005149.

2) Co/Mn covariation with Mo/Ti and d98Mo (Fig 3f)

The authors state in their response letter: “Because Mn and Co concentrations are significantly lower in seawater than in pore water, it is unlikely that bottom water influx influenced variations in the Co/Mn ratio.” However the authors also mention in the manuscript (line 301) that “Co/Mn ratio (a proxy for geochemical accumulation rate) in deep-sea sediments (Fig. 3F), likely reflecting variations in Fe-Mn (hydro)oxides mineralogy and crystal structures formed under different depositional conditions”.

There is a significant covariation between authigenic Mo enrichment (Mo/Ti ratios) and Co/Mn ratios. Especially in core GC112 which do show larger Mo/Ti variations with higher Mo/Ti ratios are leading to lower Co/Mn ratios ($R^2=0.85$). Is a systematic relationship between Co/Mn, d98Mo and Mo/Ti line with the suggested process of bottom water Mo infiltration into the sediment and subsequent cycling within the deeper sediment column or can this effect seen between Co/Mn, d98Mo and Mo/Ti be explained mineral transformation from birnessite to todorokite? (e.g. effect of mineral transformation of Ni isotopes (Fleischmann, EPSL, 2023))

Answer: We appreciate the reviewer's insightful comments and agree with Reviewer #2's perspective on the limitations of using the Co/Mn ratio in certain sedimentary contexts. Although the Co/Mn ratio is a well-established proxy in iron-manganese nodules, which are generally unaffected by early diagenesis (Manheim et al., 1998; Fleischmann et al., 2023; Zhang et al., 2024), its applicability to deep-sea sediments is less reliable. In such environments, the geochemical behavior of cobalt (Co) may be

significantly modified by diagenetic processes, as evidenced by the strong correlation observed between Co and molybdenum (Mo) concentrations ($R^2 = 0.89$) in our samples. This alteration compromises the utility of the Co/Mn ratio as a consistent indicator of sedimentation rates.

To address this limitation and ensure a robust analysis, we have utilized the sedimentation rate derived from the paleomagnetic age model of the XT19 core. This approach allows us to more accurately investigate the relationship between sedimentation rate and Mo isotope variations. Our findings indicate that the sedimentation rate has a minimal influence on Mo isotope fractionation in Pacific Ocean deep-sea sediments. Relevant content has been added in the revised manuscript (Line 261-298), as:

“The sedimentation rate is a key factor influencing the stable isotopic composition of metals in marine deposits. Recent studies by Fleischmann et al. (2023) and Zhang et al. (2024) indicate that, in the absence of post-depositional, redox-driven diagenetic alterations, accumulation rates and incorporation mechanisms account for the lighter nickel (Ni) and zinc (Zn) isotopic signatures in deep-sea sediments relative to Fe-Mn crusts. The Co/Mn ratio, derived from ferromanganese nodules unaffected by early diagenesis, serves as a proxy for this geochemical accumulation rate. Our research reveals a significant difference in the Co/Mn ratio between the XT19 and GC112 cores (Fig. 3F). If the sedimentation rate were to substantially impact the Mo isotope composition of deep-sea sediments, a corresponding discrepancy would be expected in the Mo isotope records of these cores. However, the Mo isotope profiles exhibit a relatively consistent pattern across both locations (Fig. 3F). Furthermore, the correlation between $\delta^{98}\text{Mo}$ values and the Co/Mn ratio is not statistically significant in the sediments of either core (Fig. 3F). This suggests that the sedimentation rate has a

minimal influence on Mo isotope variations in Pacific Ocean deep-sea sediments. Recognizing that the geochemical behavior of Co in deep-sea sediments may be modified by early diagenesis, which would prevent the Co/Mn ratio from accurately recording the sedimentation rate, we utilized the paleomagnetic age model-based sedimentation rate from the XT19 core to further investigate this relationship. The results indicate that sedimentation rates in the XT19 core were relatively stable, measuring approximately 175 cm/kyr in the upper section (<2.0 mbsf) and 202 cm/kyr in the lower section (>2.0 mbsf)⁴³. Despite this rate stability, notable variations were observed in the corresponding Mo isotopic compositions between these two intervals. This finding reinforces the conclusion that sedimentation rate exerts minimal influence on Mo isotope variations in these Pacific Ocean deep-sea sediments.”

Taking the GC112 core as a case study, we investigated the potential influence of the mineralogical transformation from birnessite to todorokite on molybdenum (Mo) isotope compositions. The upper section (<4.0 m) of the core exhibits lower Mo concentrations than the lower section (>4.0 m), suggesting this transformation may occur primarily in the upper sediment layers. Since manganese oxides are known to preferentially incorporate lighter Mo isotopes, the conversion of birnessite to todorokite would be expected to release these lighter isotopes into the porewater. Consequently, this process should theoretically lead to a decrease in solid-phase Mo content and a corresponding enrichment in heavier isotopes (higher $\delta^{98}\text{Mo}$ values) toward the top of the core.

However, our data reveal the opposite trend. Within the upper 4 meters of the GC112 core, $\delta^{98}\text{Mo}$ values progressively decrease toward the surface (Figure 2D), which contradicts the expected isotopic enrichment. This discrepancy indicates that the observed variations in $\delta^{98}\text{Mo}$ cannot be primarily attributed to Mo isotope fractionation during the birnessite-to-todorokite transformation. Therefore, we propose that a

continuous adsorption-fractionation process offers a more plausible explanation for the systematic evolution of Mo isotope compositions with depth in these deep-sea sediments.

It is important to note that, to our knowledge, no published studies have investigated the migratory behavior of Mo during the experimental transformation of birnessite to todorokite. Future laboratory experiments are essential to definitively characterize Mo mobility and its associated isotopic fractionation during this mineralogical change.

We hope these explanations satisfactorily address the reviewer's concerns. We are grateful for the rigorous and insightful feedback, which has helped us to significantly refine and clarify the arguments presented in our manuscript.

Reviewer #3 (Remarks to the Author):

I think the authors have fully considered my previous comments and carefully revised the manuscript. I have no more comments this time, while only a small suggestion: The figure S8 that shows comparing $\delta^{98}\text{Mo}$ values of bulk sediments and oxide phases is critical to demonstrate the burial process of Mo within the sediment profile. Thus, I recommend the authors move this figure to the main text of the manuscript, instead of the Supplementary Information.

Answer:

In the revised manuscript, we have incorporated the molybdenum isotope data from the iron-manganese oxide phase of cores GC112 and XT19 (previously in Figure S8) into Figure 2. These data are now indicated with solid symbols, and the figure caption has been updated accordingly.

Review of the revised version of the manuscript NCOMMS-25-10252A “Revised Oceanic Mo Isotope Budget from Deep-Sea Pelagic Sediments” by Wang et al. submitted to Nature Communications.

Similar to my statement in the first review of this manuscript and based on the presented data, I think the proposed scenario of Mo transport from bottom water into the sediment via diffusion and subsequent adsorption onto Fe and Mn (hydro)oxides is a possible scenario. However concerning the here presented data two major concerns stay:

1) The influence on Fe oxides on Mo and higher Mo association within the Fe (hydr)oxide leach fractions

Here three observations from the authors data argue against this significant Fe influence of Mo:

a) If authigenic Fe input has a significant affect the Mo removal, why in Mo/Ti values between two cores behave similar to Mn/Ti ratios and not Fe/Ti (Fig. 2)? In core XT19 higher Fe/Ti values compared to core GC112 would have an impact in authigenic Mo enrichment (Mo/Ti). This is not the case. In the different leached fraction of MnO and Fe (hydro) oxide phases, Mo concentration in both leach fractions covary with Mn concentration and while a negative covariation is observed with Fe! Does this show that Mo is likely bound to Mn phases?

b) As stated in my last review, Co concentration covary with Mn and Mo concentrations within the sediments presented here suggesting Mn as primary Mo host as for Co

c) Plotting the different Mo, Mn and Fe concentration data from the different leaching steps (Table S2), there is a covariation between Mn and Mo concentrations within the Mn oxide phase and eFe (hydro) oxide phase sequential extraction. Although Mn concentration are quite low in second leach step, does this show that during both leaching steps the Mo is dominantly released from Mn-phases or that these different leaching steps were not quantitatively able to fully discriminate between both phases.?

2) Co/Mn covariation with Mo/Ti and $\delta^{98}\text{Mo}$ (Fig 3f)

The authors state in their response letter: “Because Mn and Co concentrations are significantly lower in seawater than in pore water, it is unlikely that bottom water influx influenced variations in the Co/Mn ratio.” However the authors also mention in the manuscript (line 301) that “Co/Mn ratio (a proxy for geochemical accumulation rate) in deep-sea sediments (Fig. 3F), likely reflecting variations in Fe-Mn (hydro)oxidesoxide mineralogy and crystal structures formed under different depositional conditions”.

There is a significant covariation between authigenic Mo enrichment (Mo/Ti ratios) and Co/Mn ratios. Especially in core GC112 which do show larger Mo/Ti variations with higher Mo/Ti ratios are leading to lower Co/Mn ratios ($R^2=0.85$). Is a systematic relationship between Co/Mn, $\delta^{98}\text{Mo}$ and Mo/Ti line with the suggested process of bottom water Mo infiltration into the sediment and subsequent cycling within the deeper sediment column or can this effect seen between Co/Mn, $\delta^{98}\text{Mo}$ and Mo/Ti be explained mineral transformation from birnessite to todorokite? (e.g. effect of mineral transformation of Ni isotopes (Fleischmann, EPSL, 2023))